

# Horizontal and vertical structure of reactive bromine events probed by bromine monoxide MAX-DOAS spectroscopy

William R. Simpson*[1], Peter K. Peterson[2], Udo Frieß[3], Holger Sihler[4], Johannes Lampel[3,4], Ulrich Platt[3], Chris Moore[5], Kerri Pratt[2], Paul Shepson[6], John Halfacre[6, §], and Son V. Nghiem[7]

[1] Geophysical Institute and Department of Chemistry and Biochemistry, University of Alaska Fairbanks, Fairbanks, AK 99775, USA
[2] Department of Chemistry, University of Michigan, Ann Arbor, MI, USA
[3] Institute of Environmental Physics, University of Heidelberg, Heidelberg, Germany
[4] Max Planck Institute for Chemistry, Mainz, Germany
[5] Gas Technology Institute, Des Plaines, IL, USA
[6] Purdue University, West Lafayette, IN, USA
[7] Jet Propulsion Laboratory, California Institute of Technology, Pasadena, CA, USA
[§] Current address: Indiana University Southeast, New Albany, IN, USA

*Correspondence to: William R. Simpson (wrsimpson@alaska.edu)

**Abstract.** Heterogeneous photochemistry converts bromide ($Br^-$) to reactive bromine species (Br atoms and bromine monoxide, BrO) that dominate Arctic springtime chemistry. This phenomenon has many impacts such as boundary-layer ozone depletion, mercury oxidation and deposition, and modification of the fate of hydrocarbon species. To study environmental controls on reactive bromine events, the BRomine, Ozone, and Mercury EXperiment (BROMEX) was carried out from early March to mid April 2012 near Barrow (Utqiaġvik), Alaska. We measured horizontal and vertical gradients in BrO with Multiple-Axis Differential Optical Absorption Spectroscopy (MAX-DOAS) instrumentation at three sites, two mobile and one fixed. During the campaign, a large crack in the sea ice (an open lead) formed pushing one instrument package ~250km downwind from Barrow (Utqiaġvik). Convection associated with the open lead converted the BrO vertical structure from a surface-based event to a lofted event downwind of the lead influence. The column abundance of BrO downwind of the re-freezing lead was comparable to upwind amounts indicating direct reactions on frost flowers or open seawater was not a major reactive bromine source. When these three sites were separated by ~30km length scales of unbroken sea ice, the BrO amount and vertical distributions were highly correlated for most of the time, indicating the horizontal length scales of BrO events were typically larger than ~30 km in the absence of sea-ice features. Although correlation dominated most of the time, rapid changes in BrO with edges significantly sharper than this ~30km length scale episodically transported between the sites, indicating BrO events were large but with sharp edge contrasts. BrO was often found in shallow layers that recycled reactive bromine via heterogeneous reactions on snowpack. Episodically, these surface-based events propagated aloft, which required enhanced aerosol extinction aloft; however, the presence of aerosol particles aloft was not sufficient to produce BrO aloft. Highly depleted ozone (<1 nmol mol$^{-1}$) repartitioned reactive bromine away from BrO and drove BrO events aloft in cases. This work demonstrates the interplay between atmospheric mixing and heterogeneous chemistry that affects the vertical structure and horizontal extent of reactive bromine events.





## 1 Introduction

During Arctic spring, photochemical reactions on snow, ice, and aerosol particle surfaces convert halide anions that originate from sea salts, but also may recycle, to potent halogen oxidizers that cause ozone depletion events, (Barrie et al., 1988; Simpson et al., 2007b) oxidize hydrocarbons, (Jobson et al., 1994; Gilman et al., 2010; Hornbrook et al., 2016) and oxidize mercury (Schroeder et al., 1998; Steffen et al., 2008), leading to enhanced deposition of pollution to the Arctic. This chemistry depends upon the presence of sea ice, which is rapidly changing (Nghiem et al., 2007, 2013; Stroeve et al., 2012),

but understanding of environmental controls (Abbatt et al., 2012; Simpson et al., 2015) on this chemical process is lacking.

Satellite remote sensing has detected enhanced bromine monoxide (BrO) during Arctic spring (Richter et al., 1998; Wagner and Platt, 1998), but satellite-detected hotspots were sometimes not observed by *in-situ* aircraft studies (Jacob et al., 2010; Salawitch et al., 2010). Satellite-observed BrO was correlated with high winds (Jones et al., 2009; Choi et al., 2012) and

potential frost flowers (Kaleschke et al., 2004) while ground-based studies found little relationship between BrO and these proxy measurements (Simpson et al., 2007a; Halfacre et al., 2014; Peterson et al., 2016). To investigate these aspects of springtime reactive bromine chemistry, we carried out the BRomine, Ozone, and Mercury EXperiment (BROMEX) (Nghiem et al., 2013) near Barrow (Utqiaġvik), Alaska in Spring 2012. This study provided an unprecedented opportunity to investigate the relationship between sea ice and atmospheric chemical processes.


Past work has demonstrated that BrO is related to ozone abundance (Simpson et al., 2007a; Helmig et al., 2012; Peterson et al., 2015), snowpack composition and pH (Simpson et al., 2005; Grannas et al., 2007; Krnavek et al., 2012; Pratt et al., 2013), aerosol particles (Frieß et al., 2011), and atmospheric stability (Peterson et al., 2015). Also during BROMEX, Moore et al. (2014) demonstrated that sea-ice leads caused vertical mixing that brought ozone and mercury down from aloft.

Peterson et al. (2015) expanded the idea that vertical mixing was important by showing that BrO vertical profiles were affected by atmospheric stability, finding that temperature inversions were correlated to shallow BrO layers. Pratt et al. (2013) showed that halogen activation was efficient in snowpack if that snow was acidic and had enriched $Br^-/Cl^-$ ratios. Frieß et al. (2011) showed that BrO was more likely to be elevated in cases where aerosol particles were present (as indicated by aerosol extinction measurements) and suggested that aerosol production from blowing snow (Yang et al., 2010)

may have been responsible.

A particular aspect needing consideration for BrO observations is $BrO_x$ ($BrO_x$ = BrO + Br) repartitioning that occurs at low ozone mixing ratios. The two fastest reactions for the $BrO_x$ family are BrO photolysis (1) and reaction of Br with ozone (2),

$$\text{BrO} + h\nu \rightarrow \text{Br} + \text{O} \tag{1}$$

$$\text{Br} + \text{O}_3 \rightarrow \text{BrO} + \text{O}_2. \tag{2}$$

These two reactions interconvert $BrO_x$ family members, but do not change $BrO_x$ abundance. When ozone is depleted to levels ~1–2 nmol $mol^{-1}$, and for near-noon photolysis conditions, these reactions are of roughly equal rates. Therefore, when ozone goes below a few nmol $mol^{-1}$, reactive bromine ($BrO_x$) may be present but not all of that $BrO_x$ is spectroscopically visible as BrO. $BrO_x$ is instead present as Br atoms, which rapidly oxidize mercury (Holmes et al., 2006, 2009; Stephens et

al., 2012; Moore et al., 2014), affecting the fate of this pollutant. This low-ozone-induced $BrO_x$ repartitioning has been observed as decreased surface BrO during very low surface ozone periods in multiple field studies (Simpson et al., 2007a; Helmig et al., 2012; Peterson et al., 2015).





In the BROMEX study, we used Multiple AXis Differential Optical Absorption Spectroscopy (MAX-DOAS) instruments
deployed via helicopter in an upwind/downwind array to determine typical horizontal length scale and vertical structure of
BrO events. These gradient observations complement Peterson et al. (2017), who used airborne DOAS to study an episode
of reactive halogen transport. Here, we use these observations to study BrO spatial structures and the effects of sea-ice-lead
induced vertical mixing on reactive bromine.

## 2 Methods

### 2.1 MAX-DOAS spectroscopy and analysis

Bromine monoxide, BrO, and aerosol optical properties inferred from oxygen collisional dimer ($O_2$-$O_2$ or $O_4$) absorption
were measured by MAX-DOAS spectroscopy, as described in Peterson et al. (2015), who adapted the methods of Frieß et al.
(2011); see the supplemental materials for details. Three MAX-DOAS instruments of design similar to that described in
prior references (Carlson et al., 2010; Peterson et al., 2016) were used. Two of the MAX-DOAS instruments were housed
on mobile solar-powered instrument packages called "IceLanders" that were deployed by helicopter onto sea ice near the
Barrow Arctic Research Center (BARC) building, where the third MAX-DOAS instrument was located. Section 3 describes
locations of the sites.

All MAX-DOAS instruments followed a scan pattern that included below-horizon viewing elevations through the zenith
(nominally -2°, -1°, 0°, 1°, 2°, 3°, 5°, 10°, 20°, and 90° elevation angles). Spectra were analyzed relative to the zenith
spectrum in an elevation sequence to result in differential slant column densities, d$SCD$, of $O_4$, BrO, $NO_2$, and $O_3$ as
described in Peterson et al. (2015) with modifications described in the supplemental materials. At the negative elevation
angles, the physical pathlength is shortened due to viewing the ground, causing a cutoff in the d$SCD$($O_4$) as compared to
above horizon spectra. For elevation scans when the near-horizon positive elevation d$SCD$($O_4$) was greater than $2 \times 10^{43}$
molecule$^2$ cm$^{-5}$, automated software determined the below horizon half-cut elevation angle. Radiative transfer modeling
with SCIATRAN (version 3.2.5) (Rozanov et al., 2005) indicated that this half-cut elevation angle was a function of
instrument optical inlet height, occurring at -0.1° elevation for 3 meter above ground level (AGL) telescope height, which
was appropriate to the IceLander platform, and -0.3° elevation for 14 meter AGL telescope height, which was appropriate
for BARC building deployment. This optical measurement of the horizon elevation was then used to adjust the view
elevations and resulted in shifts to the observation elevation of <+/- 0.3° from the nominal elevation. These corrected
elevation angles were used in the subsequent analysis. All data are available from the NASA Earth Exchange (NEX)
platform (https://nex.nasa.gov/nex/projects/1388/).

The d$SCD$ measurements were inverted to vertical profiles of BrO mixing ratio and aerosol particle light extinction using
the University of Heideleberg Profile (HeiPro) Optimal Estimation (OE) modeling software. HeiPro version 1.4 was used
for retrieving aerosol optical properties and HeiPro 1.3 was used for BrO trace gas analysis. All data were processed as
hourly averages resulting in vertical profiles from the surface to 4 km. Only positive elevation angle (above horizon) d$SCD$
measurements were used in the OE modeling by selecting observations at elevations greater than the instrumental field of





view (0.7° FWHM). The aerosol grid consisted of 20 layers each 0.2 km thick, and the BrO grid consisted of 40 layers, each 0.1 km thick.

As discussed in Peterson et al. (2015), and following the work of Payne et al. (2009), the full grid over-represents the true information content in the retrieved vertical profiles, which was typically ~2-3 degrees of freedom for signal (DOFS) for both BrO and aerosol extinction (see supplemental Fig. S1). Therefore, the BrO amounts and vertical distributions were represented by two quantities, the Lower-Tropospheric VCD (LT-VCD), and the fraction of that LT-VCD in the lowest 200 m ($f_{200}$). We refer to this representation of the data as "information-content-based retrievals". The BrO LT-VCD is nominally the integral of the vertical profile from surface to top of model, but loses sensitivity above about 2000m. The $f_{200}$ is the partial VCD from surface to 200 m divided by the LT-VCD. For BrO retrieval, if the DOFS in either the lowest 200 m is <0.7 or the DOFS from 200 m to 2000m is <0.5, we do not report the data, assuring that only retrievals that are well constrained by the observations are used. For some figures, we calculated the average BrO mixing ratio (MR) in the 0-200 m layer, represented in pmol mol$^{-1}$. Aerosol extinction vertical profiles were retrieved by modeling d$SCD$($O_4$). The aerosol extinction profiles were then integrated from surface to 4 km to result in an aerosol optical depth (AOD, unitless), which was typically small for much of the campaign due to predominance of clear skies. Larger AOD values were likely to be caused by clouds. Supplemental Fig. S1 shows reduction in the lofted information (DOFS for BrO in the 200-2000m AGL layer) at high AOD, which is consistent with Peterson et al. (2015).

### 2.2 Ozone measurements

Surface ozone mixing ratios were measured on each IceLander platform with a modified 2B Technologies Model 205 photometric ozone monitor (Halfacre et al., 2014). These instruments had a manufacturer-specified ozone limit of detection of 1.0 nmol mol$^{-1}$. The surface ozone mixing ratio was measured in Barrow (Utqiaġvik) (McClure-Begley et al., 2014) at the NOAA Global Monitoring Division (GMD) site (71.3230°N, 156.6114°W), 2 km East of the BARC site.

### 2.3 Meteorological measurements

Barrow meteorological data was measured at the NOAA Barrow (Utqiaġvik) Airport (PABR) Automated Surface Observing System (ASOS) site, 71.286°N, 156.766°W, which is 5.5 km SW from the BARC building. Winds were measured at 10 m AGL using an ultrasonic anemometer, and temperature was measured at 2 m AGL with an aspirated thermometer. Each IceLander platform carried cup anemometers for wind speed measurements that were recorded by data loggers (Campbell Scientific). The IceLander winds were recorded at approximately 2.5 m AGL.

### 3 BROMEX field campaign and meteorological situation

The BARC instrument was located at 71.325°N, 156.668°W and was used as a point of reference for the measurements at the IceLander sites, called IL1 and IL2. Initially, both IceLanders were co-located with BARC for intercomparison purposes. On the afternoon of March 8, IL2 was deployed 27 km west of BARC (to 71.2745°N, 157.295°W), and on March 9, IL1 was deployed 36 km east of BARC (to 71.355°N, 155.668°W). At about 13:00 AKST on March 23, the sea ice on which IL2 was located broke away from the landfast ice, and IL2 entered a drift phase for the remainder of the campaign. Figure 1 shows the locations of instrument packages overlaid on a map of the sea ice near local solar noon (~13:30 AKST) on March





23, and the supplemental animation shows the temporal evolution of sea ice and motion of the packages. IL1 was recovered
from the sea ice via snowmobile near 12:00 AKST on March 27 and was operated co-located with the BARC instrument
until March 31. Following opening of the lead, IL2 drifted, reporting MAX-DOAS data until it tipped over on April 10. The
locations of IL2 and IL1 compared to the BARC instrument location are shown in the lower panel of Fig. 1.

Figure 2 summarizes the meteorological conditions during the campaign. Temperatures were cold (–15C to –35C), but
many days showed diurnal heating due to the returning sun of March/April. Winds varied from calm to 12 m s$^{-1}$, and were
approximately 10 m s$^{-1}$ at the time of ice breakoff on March 23. When winds were low, shear was evident with higher winds
at 10 m AGL (measured at the Barrow Airport site) but less wind speed closer to the surface at the 2.5 m AGL IL sites. Two
periods (March 28–April 2 and April 6–April 8) of zero reported wind speed at IL2 were likely caused by icing of the cup
anemometer at that site and are probably artifacts.

The wind direction in this period was bimodal with a predominant wind from the north through east sector and a secondary
wind peak from the west (see wind direction histograms in supplemental Fig. S2). We divided the campaign into periods of
"east" wind and "west" wind by taking the sector from 160° to 340° as "west" and from 340° through 0° to 160° as "east".
The predominant "east" wind was from the northeast, average direction 48°, standard deviation 29°, and the less-frequent
"west" wind was from the west, average direction 262°, standard deviation 34°. The design of the experiment was to have
one IL platform upwind and one downwind of BARC, and this design often worked, as evident by the wind direction being
relatively parallel to the ENE-WSW direction of the IL1-BARC-IL2 line of sites. For the majority of the deployment phase,
IL1 was upwind of BARC and IL2 was downwind. The lowest panel of Fig. 2 shows the distance of the IL platforms from
BARC. IL1 never moved significantly, but IL2's drift brought it ~250 km west, typically downwind, from BARC.
IceLander 2 MAX-DOAS observations stopped on April 10 at 17:30 AKST, when IL2 tipped 33 degrees, preventing its
optical scanner from observing the horizon.

For this analysis, we divided the campaign into four periods:
- Intercomparison: March 2 – 8, in which all instruments were compared at BARC.
- Period B1: March 8 – 18, which had no open leads and the array was deployed.
- Period B2: March 18 – 28, which covered the lead opening event when IL2 drifted west.
- Period B3: March 28th – April 10, during which time IL1 was returned to BARC and shut down and IL2 continued
  observing until tipped.

## 4 Results

### 4.1 Intercomparision of MAX-DOAS observations when co-located

To assure inter-comparability of the MAX-DOAS measurements, all hours when the IceLander instruments were at BARC
were correlated. This selection led to 24 hours of IL2-IL1 comparison and ~50 hours of BARC-IL1 comparison. Because
IL1 was recovered, pre- and post-deployment intercomparision data were determined, but IL2 was lost, so only pre-
deployment intercomparion was possible. Figure 3 shows the results of these comparisons. For BrO 0-200m mixing ratio,
we found high $R^2$ correlations between instruments of 0.92 and 0.95, intercepts <0.5 pmol mol$^{-1}$ and slopes of 1.08 and 1.29.





Typical errors ($1\sigma$) for BrO surface mixing ratio were 2-3 pmol mol$^{-1}$. For the BrO LT-VCD, $R^2$ correlations were 0.76 and 0.87 with intercept statistics within $1\sigma$ of zero and slope within $2\sigma$ of unity. Typical BrO LT-VCD errors ($1\sigma$) were ~5 x $10^{12}$ molecule cm$^{-2}$. These results demonstrate good agreement between three independent instruments and allow us to use the instruments to determine horizontal gradients.

**4.2 Gradient observations during phases B1-B3**

Figures 4-6 show atmospheric chemical observations during these three phases. During this campaign, there was a great deal of variability of BrO by all measures. The LT-VCD varied ~0 to ~8 x $10^{13}$ molecule cm$^{-2}$ and the fraction in the lowest 200m ($f_{200}$) varied the full possible range, 0 to 1. Generally, period B3 had lower $f_{200}$ values, and period B2 lacked very high column events.

During period B1, BrO at the three sites followed the same behavior. Large changes such as the precipitous drop in BrO LT-VCD from >6 x $10^{13}$ molecule cm$^{-2}$ on March 15 at sunrise down to near zero values in the late morning happened at all three sites. This change appeared to be at least partially due to low-ozone-induced BrO$_x$ repartitioning at the surface, as discussed in Section 5.3. Decreasing AOD was also observed at this time, which was probably the result of an airmass change (e.g. frontal passage). The vertical structure ($f_{200}$) also agreed very well between sites. There were some time shifts of up to ~2 hours between sites, which was consistent with the corresponding transport time, but the sites generally followed the same pattern even if they were shifted in time by an hour or two.

Period B2 started with consistent meteorology from period B1, although the wind increased from March 18th up until the ice breakoff event on March 23. The BrO LT-VCD during this period was lower than its peak earlier in the campaign, and very shallow (e.g. $f_{200} > 0.5$) events were observed. The association of shallow BrO layers with small column density was noted by Peterson et al. (2015) and was interpreted as a result of poor vertical mixing preventing propagation of surface-based BrO aloft. During the static phase (prior to ice break off and IL2 motion) of the study (March 8 to 23), correlations between BrO measurements at BARC and IL2 compared to IL1 were still high despite horizontal separation (see supplemental Fig. S3). The BARC-IL1 LT-VCD correlation $R^2$ value was 0.84 and the IL2-IL1 correlation was $R^2 = 0.79$. The surface BrO mixing ratios were similarly correlated, with $R^2$ values of 0.85 and 0.81, respectively. These correlation coefficients were similar to the co-located period, despite separation between sites of 36 km (BARC-IL1) and 63 km (IL2-IL1) respectively. Supplemental Fig. S4 shows histograms of LT-VCD differences between sites. This analysis shows that the probability of having a difference with an absolute value less than $10^{13}$ molecule cm$^{-2}$ ($2\sigma$ of the BrO LT-VCD measurement error) is 87% for IL1-IL2, and 83% for IL1-BARC, again indicating that, at most times without open sea-ice leads, strong spatial gradients in BrO are not observed.

Figures 4 and 5 also demonstrate that ozone at the three locations was highly correlated before the lead opening event on March 23. Generally, before lead opening, surface ozone mixing ratios were low (< ~15 nmol mol$^{-1}$), but when the wind speed increased (March 10, 11 and surrounding the lead opening on March 23), ozone mixing ratios increased, consistent with ozone downward transport from aloft as reported by Moore et al. (2014). We also observed reduced wind shear between the Barrow measurements (at 10m AGL) and the IL platforms (2.5m AGL) during the higher wind periods, consistent with reduced stratification of air near the surface.





Upon lead opening on the afternoon of March 23, changes to the vertical structure of BrO appeared. The most downwind site, IL2, was within a surface-based cloud formed by the open lead and the AOD increased significantly at that location as compared to the other sites. This cloud precluded BrO observations at IL2 until the next day (March 24), on which a gradient in $f_{200}$ developed between IL1 (upwind), BARC (middle), and IL2 (downwind), with a shallower BrO distribution (higher $f_{200}$ values) at the upwind site and a more vertically mixed behavior (lower $f_{200}$ values) at IL2. Note that the surface

ozone mixing ratio was high enough (more than a few nmol mol$^{-1}$) that repartitioning of $BrO_x$ at the surface was not responsible for these lower $f_{200}$ values at IL2. This gradient in BrO vertical structure persisted until the morning on March 26, when the three sites appeared to be in different air masses (as indicated by different ozone mixing ratios at the three sites, in contrast to their prior highly correlated behavior). This change in ozone at Barrow (BARC) on March 26 was noted by Moore et al. (2012) as a reduction in vertical mixing due to re-freezing of previously open leads upwind of BARC.


On the afternoon of March 27, IL1 was recovered and resumed operation at BARC. During this post-deployment co-location period, IL1 and BARC agreed well (Fig. 6). On March 28 (Fig. 6), there was a different vertical profile at IL2 as compared to BARC, with a surface-based event on the sea ice at IL2 that had higher $f_{200}$ values than BARC. During the downwind drift period (B3), the highly correlated behavior observed prior to lead opening on March 23 was replaced with

more variability between IL2 and BARC. However, the daily-timescale values of all quantities appear similar between these sites despite the large distance (~130 to 260 km) between sites.

### 4.3 Selected cases

Figures 7 and 8 show altitude / time profiles of aerosol extinction and BrO mixing ratio, respectively, on the selected days of March 9, 15, 16, and 22-24. These cases were selected to have similar meteorological conditions, with winds from the

northeast, moderate to low AOD (except for around lead opening event, particularly at IL2, on the afternoon of March 23), and cold temperatures (-25°C to -35°C). There was generally good agreement between the three sites, in agreement with the information-content-based analysis shown in Figs. 4-6. These altitude / time profile presentations are informative despite the fact that they over-represent the vertical information, particularly aloft, where the averaging kernels show that the vertical resolution is broadened significantly (Frieß et al., 2011; Peterson et al., 2015). It is likely that some subtle differences aloft

are simply due to lack of vertical resolution, but the consistent features between sites are likely well represented in these profile plots.

### March 9

This case shows a vertically thick surface-based aerosol layer, with $\log_{10}$(extinction) > −1 up to ~1 km. During this day, surface ozone was sufficiently low (<1 nmol mol$^{-1}$) after noon to cause $BrO_x$ repartitioning at the surface, which is evident

by reduced values of $f_{200} < 0.1$ (Fig. 4). The time profile plots show that BrO was not present at the surface, but the peak mixing ratio moved aloft. There was a moderate decay in the LT-VCD on this day, but the presence of surface $BrO_x$ repartitioning did not eliminate BrO.



**March 15**

This day had dramatic BrO changes with nearly temporally coordinated behavior at all three sites. Figure 4 shows that in the morning, there was high BrO LT-VCD ($>7 \times 10^{13}$ molecule cm$^{-2}$), which declined to near zero values ($<1 \times 10^{13}$ molecule cm$^{-2}$) at noon, and then recovered moderately ($\sim2 \times 10^{13}$ molecule cm$^{-2}$). The morning vertical distribution of BrO showed $f_{200} = 0.3$, which decreased to low values, consistent with surface-based BrO$_x$ repartitioning at low ozone levels, and then $f_{200}$ increased again to $>0.5$ late in the afternoon, indicating a surface-based BrO layer. The BrO profiles (Fig. 8) show a relatively thick BrO layer (to 1 km) in the morning that decayed to zero at noon and then built a shallow event in the afternoon. Aerosol extinction on this day was at relatively high values in the morning, but decreased to low levels (log(extinction) $< -1.2$), particularly above the first 200 m AGL in the afternoon.

**March 16**

This day had a shallow surface-based BrO event with $f_{200}$ values between 0.4 and 0.8. Sufficient ozone was present at the surface to prevent repartitioning of BrO$_x$. The aerosol profiles show that there was very little aerosol extinction aloft, and only small amounts in the lowest few hundred meters. Figure 7 shows some evidence of a lofted aerosol particle layer, but that layer was decoupled from the surface aerosol layer and was not associated with BrO enhancements.

**March 22**

This case, which was the day before the lead opening event, shows an interesting contrast to March 16. There was again a surface-based BrO event, with $f_{200} > 0.6$, which was slightly more shallow than March 16. However, the aerosol extinction was both higher in magnitude on this date and was distributed much more aloft than on March 16. Figure 7 demonstrates that the aerosol layer descended throughout the day and seemed to be overlapping the surface layer. However, despite the presence of aerosol particles aloft, Fig. 8 shows that BrO does not appear aloft (as it had on March 9 and March 15 morning), as discussed in Section 5.4.

**March 23**

This was the day of the lead opening event. All three sites began with a shallow BrO event in the morning. There was moderate aerosol extinction, mostly based at the surface, but extending aloft. At the time of the lead opening event, the aerosol extinction at IL2 (downwind) went high, $>1$ km$^{-1}$, in the lowest 400 – 600 m AGL, consistent with that instrument being within the convective lead cloud. Unfortunately, the lead cloud prevented BrO LT-VCD or $f_{200}$ from being observed at IL2, but observations become valid at all three sites on the next day.

**March 24**

This case shows the next-day response of BrO to this lead opening event. Downwind of the open and re-freezing lead, IL2 observed a decrease in BrO mixing ratio at the surface (Fig. 8) and a broadening of the BrO vertical profile to greater heights. Unlike most times earlier in the campaign, BrO (Fig. 7) and aerosol extinction (Fig. 8) show gradients between the sites, as was discussed using BrO LT-VCD and $f_{200}$ earlier in this section. The downwind IL2 site had high aerosol extinction in a thick ($>400$m) surface based layer, which decayed in the afternoon. Ozone was high ($\sim30$ nmol mol$^{-1}$) at all sites, eliminating BrO$_x$ repartitioning as a cause for this difference.



## 5 Discussion

### 5.1 BrO measurements were highly correlated on ~30 km length scales in the absence of leads

During the pre-lead-opening period (before March 23), Figs. 4 and 5 show that measurements at the three sites correlate
despite physical separation between sites of ~30 km, and even ~60 km from IL2 to IL1. The correlation (Fig. S3) and $R^2$
values of these separated measurements in this period are quite similar to the times when the instruments were co-located at
BARC (Fig. 3). It is evident from examination of the time series data (Figs. 4 and 5) that some changes in BrO occured at
one site before another, with temporal shifts of a couple hours. This type of temporal shift would have decreased the hourly
correlation coefficient, and was likely responsible for some of the reduction of $R^2$ between BrO correlations for the
deployed site locations and the co-located sites. The high correlation of measurements separated by length scales similar to
satellite pixel dimensions (Richter et al., 1998; Wagner and Platt, 1998; Begoin et al., 2010; Choi et al., 2012; Sihler et al.,
2012) is an important finding that generally indicates that satellite-based BrO observations are likely to represent horizontal
spatial features effectively. Variability of BrO in the stratosphere (Theys et al., 2009; Salawitch et al., 2010) or free
troposphere (Theys et al., 2011; Choi et al., 2012) could affect this conclusion, but one would expect less horizontal
inhomogeneity aloft because of lack of small-scale features such as leads or topography, which are only present at ground
level. Although we observe that the general behavior of BrO is high correlation despite spatial separation, transport events
that have gradients significantly sharper than satellite lengths scales are clearly evident in the data. For example, on March
13, Fig. 4 shows time-staggered changes in BrO LT-VCD and $f_{200}$. Peterson et al. (2017) used airborne DOAS to study the
March 13 case and observed a very steep BrO gradient that transports with the wind, clearly indicating that features on the
edges of air masses are sharp. Therefore, we interpret the BrO distribution as being large regions of relatively consistent
BrO on >30km length scales with sharp contrasts at their edges that are much smaller than satellite length scales.

### 5.2 Snowpack-based BrO events were common during BROMEX

Many of the BrO events that occurred during BROMEX were ground-based with a high fraction of the BrO LT-VCD in the
lowest 200m (large $f_{200}$). Peterson et al. (2015) showed that shallow events are associated with stable meteorological
conditions, which predominated during much of this campaign, particularly before the lead opening event. These shallow
events are consistent with a snowpack source of reactive bromine (Simpson et al., 2007a; Pratt et al., 2013). Reactive
bromine is relatively short lived due to termination reactions, which often lead to HBr or HOBr bromide reservoirs (Platt
and Hönninger, 2003). However, these reservoir species can recycle to reactive halogens through heterogeneous chemical
reactions. Fan and Jacob (1992) proposed that heterogeneous reaction (3) on aerosol surfaces was a critical step for
recycling reactive bromine and activating particle-bound bromide (Br$^-$) to reactive bromine after photolysis of Br$_2$.

$$HOBr + H^+ + Br^- \rightleftarrows Br_2 + H_2O \qquad (3)$$

Subsequent laboratory studies have demonstrated that reaction (3) is efficient on saline liquid, ice, and aerosol particle
surfaces (Fickert et al., 1999; Huff and Abbatt, 2000, 2002; Wachsmuth et al., 2002; Abbatt et al., 2012; Wren et al., 2013).
Because heterogeneous chemistry is required for reactive bromine recycling, we interpret these surface-based events as
recycling reactive bromine on snowpack surfaces.

As evident from the case studies on March 16 and 22, shallow surface-based BrO events can have different relationships to
aerosol vertical structures. On March 16, there was little aerosol extinction aloft, implying low aerosol surface area density,





which would have slowed heterogeneous recycling on lofted aerosol particles. One might interpret the lack of aerosol
particles aloft as causing the event to be surface based. However, on March 22, there is significantly more aerosol extinction
detected aloft, but the event remained based at the surface. Potential reasons for BrO not being observed aloft on March 22
despite the presence of aerosol particles could be lack of vertical mixing due to meteorological inversions (Peterson et al.,
2015), which were common during the campaign. Specifically, on March 22, the meteorological sounding balloon launched
from Barrow (Utqiaġvik) at 15:00 AKST showed an inversion with $dT/dz$ of +15 K km$^{-1}$ in the lowest 200m AGL. An
alternative hypothesis for the lack of reactivity on the lofted aerosol on this date could be that the particles had a chemical
composition that was not conducive to halogen release. For example, if the particles didn't contain bromide (Br$^-$), reaction
(3) would not occur. Laboratory (Fickert et al., 1999; Huff and Abbatt, 2002; Abbatt et al., 2012; Wren et al., 2013) and
field (Pratt et al., 2013) studies indicate that acidic pH is also required for reaction (3), adding another potential reason.
Another alternative could be that the aerosol size distribution consists of larger particles for which diffusion limits gas-
surface reaction rates.

### 5.3 Low-ozone-induced BrOx repartitioning affected BrO vertical profiles

Past considerations of reactive bromine chemistry has indicated that $BrO_x$ partitioning between Br atoms and BrO can be an
important control on BrO abundance (Simpson et al., 2007a; Helmig et al., 2012). The low ozone mixing ratios observed
here (often <1–2 nmol mol$^{-1}$) controlled surface $BrO_x$ partitioning and reduced BrO abundance, and thus affected the
vertical distribution of BrO. Low BrO concentrations would also have reduced the formation of HOBr, which is necessary
for "bromine explosion" events (Wennberg, 1999; Lehrer et al., 2004) that recycle $BrO_x$ via reaction (3). Through reduced
heterogeneous recycling, $BrO_x$ would have decayed over time as termination reactions (e.g. Br + H$_2$CO) occurred. On
March 9, ozone levels began the day above this threshold, but soon decayed below the threshold, and BrO at the surface
decayed to zero (Fig. 8). This repartitioning effect reduces the $f_{200}$ value to <0.1 (Fig. 4), and BrO exists only aloft in the
afternoon (Fig. 8). On March 9, the reactive bromine aloft was apparently generated at the surface and moved aloft where it
recycled on aerosol particle surfaces.

March 15 also had low ozone values and what started as a very intense BrO event in the morning decayed to near zero LT-
VCD at noon. At noon, the vertical structure of BrO became lofted ( $f_{200}$ <0.1; Fig. 4), but aerosol extinction aloft was
smaller (Fig. 7 and see Section 5.4) and BrO did not propagate aloft after noon (Fig. 8). Interestingly, on March 15, a
shallow ($f_{200}$ > 0.5) post-noon BrO column appeared (Fig. 8), potentially enabled by decreased afternoon photolysis rates
and an increase in O$_3$ in the late afternoon that repartitioned $BrO_x$ back towards BrO.

BrO$_x$ repartitioning may also have been responsible for low surface BrO levels and low $f_{200}$ values on many of the days
during this campaign. Peterson et al. (2016) found that this period (spring 2012) had particularly low surface ozone, and
Oltmans et al. (2012) showed that March ozone depletion event (surface O$_3$ < 10 nmol mol$^{-1}$) probability has been
increasing over the 38-year period from 1972—2010. Therefore, the prevalence of BrO$_x$ repartitioning in the BROMEX
data set may not be representative of average behavior and warrants further climatological investigation through analysis of
larger data sets.





### 5.4 Aerosol extinction aloft was necessary but not sufficient for BrO to be present aloft

The cases presented in Section 4.3 and discussed above found that shallow BrO events sometimes occurred with little aerosol aloft (March 16), and at other times with significantly more aerosol aloft (March 22). When $BrO_x$ repartitioning affected surface BrO, sometimes the BrO event migrated aloft in the presence of significant aerosol loading (March 9), but sometimes BrO decayed both at the surface and aloft (March 15). In all of these cases, BrO only propagated aloft into layers with $\log_{10}$ (aerosol extinction) $> \sim -1$, meaning aerosol extinction coefficient ($k_{ext}$) $> \sim 0.1$ km$^{-1}$. Aerosol extinction is related to aerosol surface area density by $k_{ext} = Q_{ext} * SA / 4$, where $Q_{ext}$ is the extinction efficiency, which maximizes for submicron particles at a value close to 4, and $SA$ is the surface area density. Assuming maximal $Q_{ext} = 4$, we can convert $k_{ext}$ to a surface area density and find $k_{ext} = \sim 0.1$ km$^{-1}$ would indicates $SA = \sim 100$ micrometer$^2$ cm$^{-3}$ appears necessary for BrO to propagate aloft.

In the absence of diffusion limitations (e.g. typically for submicron particles), the rate of a heterogeneous reaction is: $k_{het} = 1/4 \, c \, \gamma \, SA$, where $c$ is the average velocity of the gas, and $\gamma$ is the reaction probability. Wachsmuth et al. (2002) indicate that heterogeneous uptake of HOBr on aerosol particles is limited by accommodation, and has the value $\gamma = 0.6$. At 100 micrometer$^2$ cm$^{-3}$, and thermal velocity of HOBr at 253K (-20°C), $c = 255$ m s$^{-1}$, thus $k_{het} = 0.0038$ s$^{-1}$, corresponding to an $\sim 4$ minute HOBr lifetime. Thompson et al. (2015) indicate the photolysis rate, $J$(HOBr) $= 0.0023$ s$^{-1}$ for springtime Barrow (Utqiaġvik) conditions, so this surface area density results in a heterogeneous reactivity rate that competes with HOBr photolysis. Photolysis of HOBr cycles reactive bromine and destroys ozone, but does not increase the reactive bromine pool. On the other hand, reaction (3) forms $Br_2$, and upon $Br_2$ photolysis results in two reactive bromine species from the one BrO radical that formed HOBr (Wennberg, 1999; Platt and Hönninger, 2003). Thus, for bromine to "explode", heterogeneous reactions must occur fast enough to compensate for reactive bromine losses (e.g. termination reactions such as Br + $H_2CO$). The observational threshold found in this study appears to be sufficiently high to allow heterogeneous recycling of $BrO_x$ to compete with $BrO_x$ loss. Therefore, it appears reasonable that current understanding of bromine chemical kinetics is in agreement with this observed aerosol optical extinction threshold (aerosol extinction > 0.1 km$^{-1}$) required for BrO to exist aloft.

Although we found that increased aerosol aloft was necessary for BrO to be found aloft, there were cases in which BrO remained ground based despite significant aerosol extinction above. For example, on March 22, there was significant aerosol extinction aloft (Fig. 7), but BrO did not show signs of migrating aloft (Fig. 8). As discussed in section 5.2, hindered vertical mixing was a likely cause of BrO remaining near the ground, but we cannot rule out scenarios where the aerosol particles didn't have the right chemical composition to recycle reactive bromine. Therefore, we find that aerosol aloft is necessary for BrO to be present aloft, but it is not sufficient to always cause BrO to propagate vertically when enhanced aerosol extinction is present. Peterson et al. (2017), used airborne DOAS to study the case on March 13 and found that a reactive bromine plume propagated with the wind and was maintained by heterogeneous chemistry on aerosol particles, complementing the detailed cases explored in the present study.

### 5.5 Sea-ice-lead-associated convection affected the BrO vertical profile

On March 23, after the opening of the sea ice lead, and on the following days (March 24 and 25) at the upwind IL1 site, BrO was present in a shallow layer ($f_{200} > 0.5$) with moderately enhanced ($2 \times 10^{13}$ molecule cm$^{-2}$) LT-VCD (Fig. 5). However,





the sites near to and downwind of the lead (BARC and IL2), exhibited decreased $f_{200}$ values as compared to the upwind site, as would be expected by vertical entrainment of reactive-bromine-poor air from above the shallow boundary layer, and

mixing of surface air aloft. Consistent with decreased $f_{200}$, Fig. 8 shows this vertical mixing decreased the BrO surface mixing ratio downwind of the lead at IL2. As opposed to the clear surface mixing ratio decrease after lead opening, the BrO LT-VCD (Fig. 5) does not show strong differences between sites along the transport direction.

To further explore the effect of the lead opening even, Fig. 9 shows the average and variability of BrO LT-VCD and vertical

distribution ($f_{200}$) for March 24 – 25, which were the two days following the lead opening event. During these two days, IL2 was downwind of a large area of re-freezing sea water, 71 – 106 km downwind of BARC. The typical wind speed was ~5 m s$^{-1}$ coming from 70°, nearly parallel to the BARC – IL2 direction, a 4-6 hour transport time. Figure 9 (left panel) demonstrates that the BrO column peaks at the middle (BARC) site and not the most downwind (IL2) site. On average, the BrO LT-VCD at IL1 (upwind) was somewhat (28%) smaller than BARC, and IL2 was slightly (5%) smaller than BARC. A

paired t-test shows that the LT-VCD was statistically significantly larger at both BARC and IL2 than it was at IL1, but that the BARC and IL2 sites were statistically indistinguishable. Figure 9 (right panel) demonstrates a clear trend in the $f_{200}$ metric of BrO vertical distribution, with a shallower surface layer (larger $f_{200}$) at the upwind IL1 site trending towards a more vertically mixed (smaller $f_{200}$) at the downwind IL2 site. All sites are statistically significantly different from each other for $f_{200}$. These observations show that the open lead's primary influence was to alter the vertical distribution of BrO,

increasing its vertical extent, but the lead only slightly affected the BrO column density on the timescale of transport between these sites (2-6 hours).

The presence of the open and re-freezing lead could have had multiple effects on aerosol particles and BrO. Wind blowing across the lead is likely to produce sea salt aerosol particles (May et al., 2016), which could be lofted in the convective

environment of the lead cloud. The lead is also refreezing between BARC and IL2 during this period, and that new ice is likely covered with frost flowers, which have been proposed as either a direct source of reactive halogens (Rankin et al., 2002) or a source of sea salt aerosol particles (Kaleschke et al., 2004) that could subsequently produce/recycle reactive bromine via Reaction (3). Figure 7 shows the lead opening event produced high extinction (>1 km$^{-1}$) through ~600m altitude on March 23, and this aerosol / cloud layer persisted into the morning of March 24. Note that MAX-DOAS

measures aerosol extinction by attenuation of $O_4$ absorption path length, so both submicron aerosol particles and supermicron particles and solid/liquid water droplets in a cloud will increase the aerosol extinction. Interestingly, the aerosol extinction at IL2 on the afternoon of March 24 seems lower than the other sites, potentially due to enhanced scavenging by the humid lead cloud environment. On March 25, Fig. 5 shows that the AOD at BARC and IL2 went below 0.2, also potentially caused by scavenging and/or reduced wind speeds (Fig. 2). Overall, the response to the lead opening of

aerosol extinction, as measured by MAX-DOAS, was a peak in aerosol extinction that corresponded with the highest winds and most downwind site (IL2) and then lower aerosol levels as winds slowed and the lead refroze. Following the passage of the lead cloud, which cleared around noon on March 24, aerosol extinction (Fig. 7) does not appear to be enhanced at the downwind site (IL2) as compared to the other sites, potentially indicating that open water and/or frost flowers between BARC and IL2 are not efficient aerosol particle sources, at least at this wind speed (which dropped to ~4 m s$^{-1}$).




With regard to BrO observations on the two days following lead opening, Fig. 9 showed that BrO LT-VCD increased 28% from IL1 to BARC, but decreased 5% from BARC to IL2. The origin of the moderate increase from IL1 to BARC is not clear, but Peterson et al. (2015) found that shallower layers (as are observed at IL1 as compared to BARC) are correlated with lower LT-VCD, so the deepening of the BrO layer, and heterogeneous reactions on lofted aerosol particles, between

445 IL1 and BARC could be responsible for that moderate increase in BrO. MODIS images (see supplemental animation) show that the lead between BARC and IL2 is refreezing in this period and given the cold temperatures (-23°C to -32°C), humidity from the open water, and presence of re-freezing sea ice, it is highly likely that the area between BARC and IL2 contained frost flowers, which have been suggested to be a source of reactive bromine (Rankin et al., 2002; Kaleschke et al., 2004). However, we see a small decrease in BrO from BARC to IL2, which argues against frost flower being a direct source of

450 reactive bromine.

### 5.6 Relationship of these findings to prior studies

These data show that vertical mixing deepens the BrO through lead-induced vertical mixing, and may over time increase the column density of BrO through heterogeneous chemistry on lofted aerosol particles. McElroy et al. (1999) observed a large tropospheric BrO column from high-altitude aircraft and associated this column with sea-ice-lead induced vertical mixing,

455 consistent with our observations. Satellite-based spectrometers detect the total (tropospheric + stratospheric) BrO column density, which can be corrected for stratospheric BrO (Theys et al., 2011; Choi et al., 2012; Sihler et al., 2012) to give a tropospheric VCD, but satellite sensors cannot determine vertical profiles of BrO and may not observe shallow BrO events, which were common during BROMEX. Thus, in-situ observations would indicate differing environmental controls for halogen activation than satellites would have indicated. Just these differences in environmental controls have been noted in

460 the literature, depending upon the type of sensor (satellite versus ground based) that was used to quantify halogen activation (BrO). Jones et al. (2009) and Yang et al. (2010) have found that satellite-detected BrO is associated with high winds that would decrease atmospheric stability and thus cause vertical mixing much like the lead-induced mixing in this example and increase visibility of BrO from space.

465 The finding that BrO is not increased downwind frost flowers is in agreement with measurements of their chemical composition, which is not conducive to reactive bromine production (Kalnajs and Avallone, 2006; Abbatt et al., 2012; Pratt et al., 2013). It is possible that the "Potential Frost Flowers" (PFF) metric (Kaleschke et al., 2004) devised to diagnose regions of frost flower formation, which involved a combination of open water and cold temperatures, could have been diagnosing spatial regions where lead-induced vertical mixing was occurring (Nghiem et al., 2012), which are correlated

470 with higher BrO LT-VCD (Peterson et al., 2015). Therefore, the correlation of PFF with satellite-observed BrO could be expected, not because frost flowers are directly responsible for halogen activation, but because vertical mixing associated with the PFF proxy enhances the thickness of the BrO layer.

Airborne observations of BrO were targeted during the NASA ARCTAS field mission to locations of high satellite-detected

475 BrO column densities, but little in-situ BrO was found (Jacob et al., 2010). This finding is again consistent with our observations – regions of high column BrO are vertically mixed, leading to lower in-situ mixing ratios of BrO, thus less detectable by aircraft in-situ techniques. Recent studies (Jones et al., 2009; Begoin et al., 2010; Choi et al., 2012) have found that mesoscale cyclonic storms that have high winds that destabilize the otherwise stable Arctic atmosphere are





associated with multi-day satellite-remote-sensed BrO transport events, again in agreement with the finding that vertical mixing enhances the BrO column density (Peterson et al., 2015).

## 6 Conclusions

Analysis of time series of the BrO lower tropospheric VCD (LT-VCD) and fraction of BrO in the lowest 200m ($f_{200}$) at Barrow (Utqiaġvik) showed the following results: When a large sea-ice lead opened and the ocean refroze, the vertical distribution of BrO was affected, but no significant increase in BrO LT-VCD was observed between BARC and IL2, which

was downwind of the re-freezing lead, providing a counterexample to the hypothesis that frost flowers growing on sea ice are a direct source of BrO. Measurements of BrO LT-VCD and $f_{200}$ were highly correlated on ~30 km length scales when there were no sources of vertical mixing (e.g. open leads) in the intervening area. During the BROMEX period, which was characterized by clear skies and cold temperatures that enhance vertical stability, shallow surface-based BrO events were common. Repartitioning of $BrO_x$ due to low ozone levels caused low surface BrO mixing ratios, and depending upon

whether reactive bromine recycling was efficient aloft, BrO either shifted to higher altitude, becoming a lofted layer, or BrO decreased through the column. Aerosol extinction aloft was necessary but not sufficient for BrO to be present aloft. An aerosol extinction larger than 0.1 km$^{-1}$ appeared necessary for maintaining BrO aloft.

These observations highlight spatial features of BrO events in the Arctic and their relationship to aerosol extinction. Vertical

atmospheric structure (stability) is a critical control on the nature of reactive bromine events, with typical "inverted" temperature profiles holding BrO close to the snowpack (Frieß et al., 2011; Peterson et al., 2015), where halogen activation reactions occur (Pratt et al., 2013). Punctuated vertical mixing events, either by sea-ice-lead-induced convection or by high winds associated with storms dilute the surface mixing ratio, but these more vertically mixed events are correlated with enhanced BrO column density. These detailed observations resolve many past controversies with respect to halogen

activation in the Arctic. The Arctic sea ice pack is thinning and multi-year ice is being replaced by seasonal first-year ice (Maslanik et al., 2011), which has been predicted to increase the occurrence of leads and has been predicted to have many further implications (Bhatt et al., 2014). Moore et al. (2014) showed that ozone and mercury are brought down from aloft during these lead events. In this work, we showed that reactive bromine was brought up from the near surface to a thicker layer by a lead-induced mixing event. These two factors should increase the overlap of mercury with reactive bromine and

thus the oxidation and deposition of mercury to the Arctic. Predictions of increased sea-ice leads would thus be expected to increase the amount of toxic mercury deposition, and have a greater impact on Arctic free-tropospheric $O_3$

**Acknowledgements:** This work was supported by the National Aeronautics and Space Administration (NASA) Cryospheric
Sciences Program (CSP), using methods supported by the National Science Foundation under grant (ARC-1023118), and by the University of Alaska Fairbanks and the Desert Research Institute. The research at the Jet Propulsion Laboratory, California Institute of Technology, was supported by the NASA CSP and by the NASA Atmospheric Composition Program. The research at University of Michigan was supported by the National Aeronautics and Space Administration (NASA) Earth Science Research Program (NNX14AP44G). We thank Bristow Helicopters and Umiaq for field logistic assistance and the National Oceanic and Atmosphere Administration (NOAA), Global Monitoring Division for the Barrow (Utqiaġvik)
Observatory data. We thank Alexei Rozanov from IUP Bremen for providing the SCIATRAN radiative transfer model. We



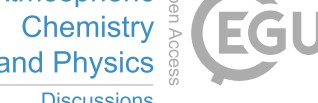

thank Todd Valentic (SRI), Steven Walsh (UAF), Don Perovich (CRREL), and Matthew Sturm (UAF) for assistance in deploying instruments and telemetry of data. Data from this project are accessible at: https://nex.nasa.gov/nex/projects/1388/






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





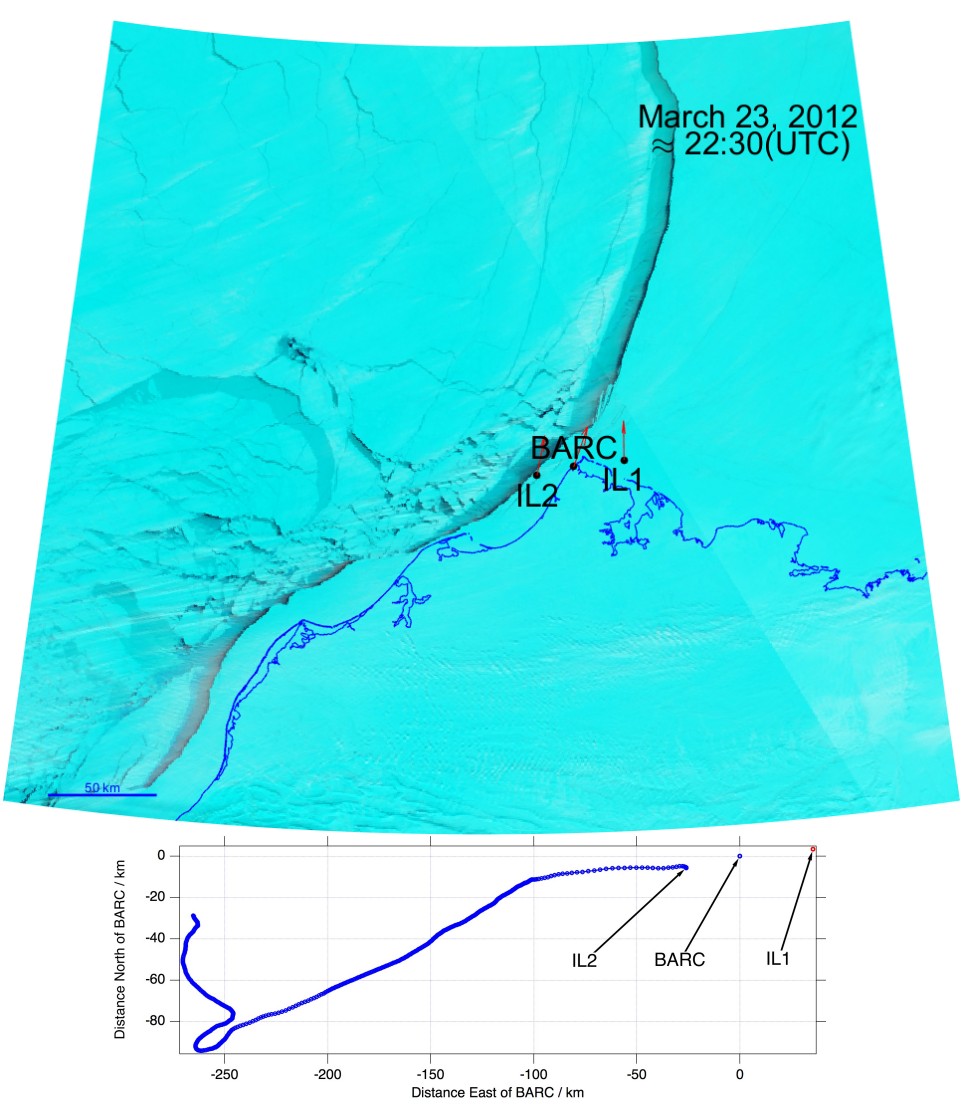


**Figure 1: The top panel shows the locations of IL1, BARC, and IL2 instruments overlaid on the March 23, 2012 sea ice map from daily RGB composite 250m-resolution MODIS images of ice conditions using the 7-2-1 bands (2,105–2,155 nm, 841– 876 nm and 620–670 nm wavelengths) from the NASA Aqua satellite. In this composite, sea ice is light blue, open-water is black, and clouds are white. The red arrows show the MAX-DOAS viewing direction, over which BrO was averaged. The wind was from the northeast (can be seen by thin "cloud streets") and was pushing the lead open and causing IL2 to break away from shore-fast ice and drift westward. IL1 was the most upwind instrument, BARC in the middle, and IL2 downwind. The lower panel shows the location of the two mobile platforms (IL1 and IL2) relative to BARC building during the drift phase. See Fig. 2 for drift distance versus time.**




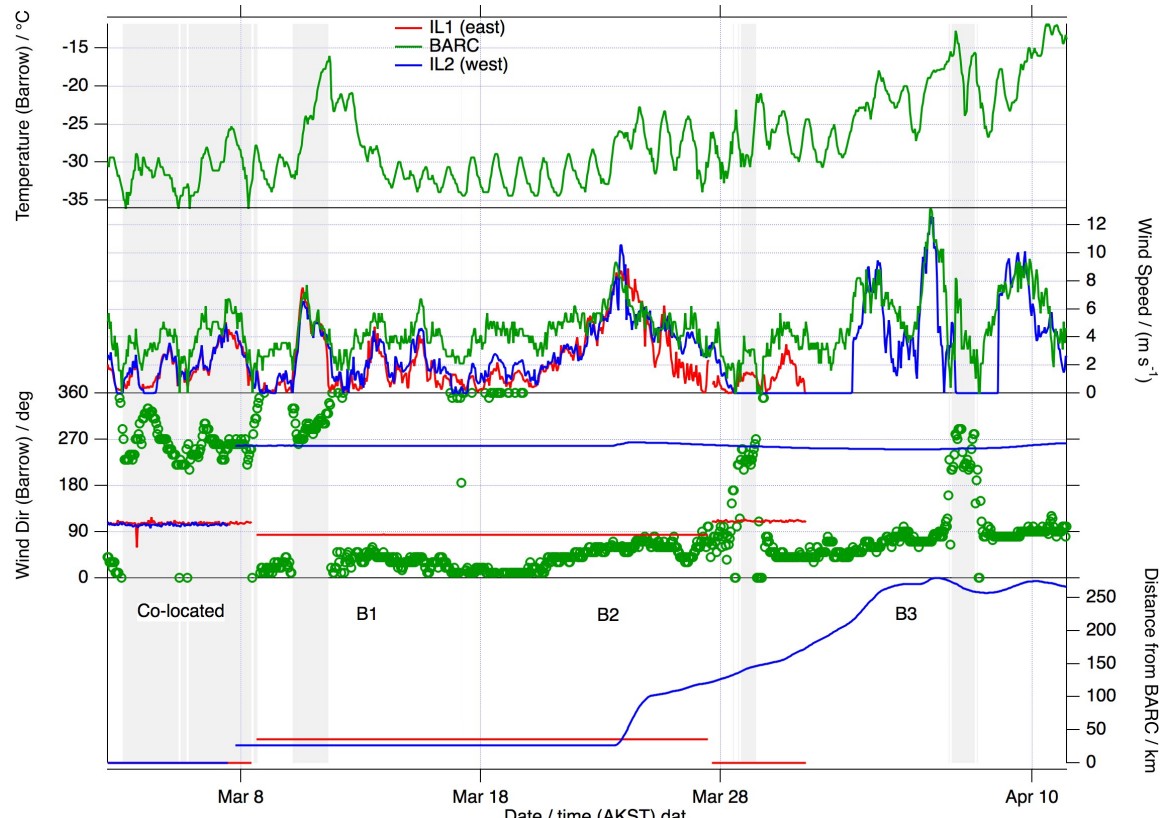

**Figure 2: Meteorological data from the BROMEX campaign and drift information of IL1 and IL2. The top panel shows Barrow
        (Utqiaġvik) temperature (2m AGL) measured at the NWS-AWOS site. The second to top panel shows wind speed at all three
        sites, but the Barrow winds were measured at 10m AGL, while IL1 and IL2 were measured at 2.5m AGL. The third panel shows
        wind direction at Barrow (Utqiaġvik, green circles) and the direction of the IL platforms from the BARC building. The wind
        direction was bimodal (Fig. S2), and west winds are plotted with a shaded background and east winds without shading. The
bottom panel shows the distance of the IL platforms from BARC. Both IL platforms started at BARC and were deployed on
        March 8 and 9. On March 23, IL2's sea ice broke away from the land starting its drift phase, and on March 27, IL1 returned to
        BARC.**





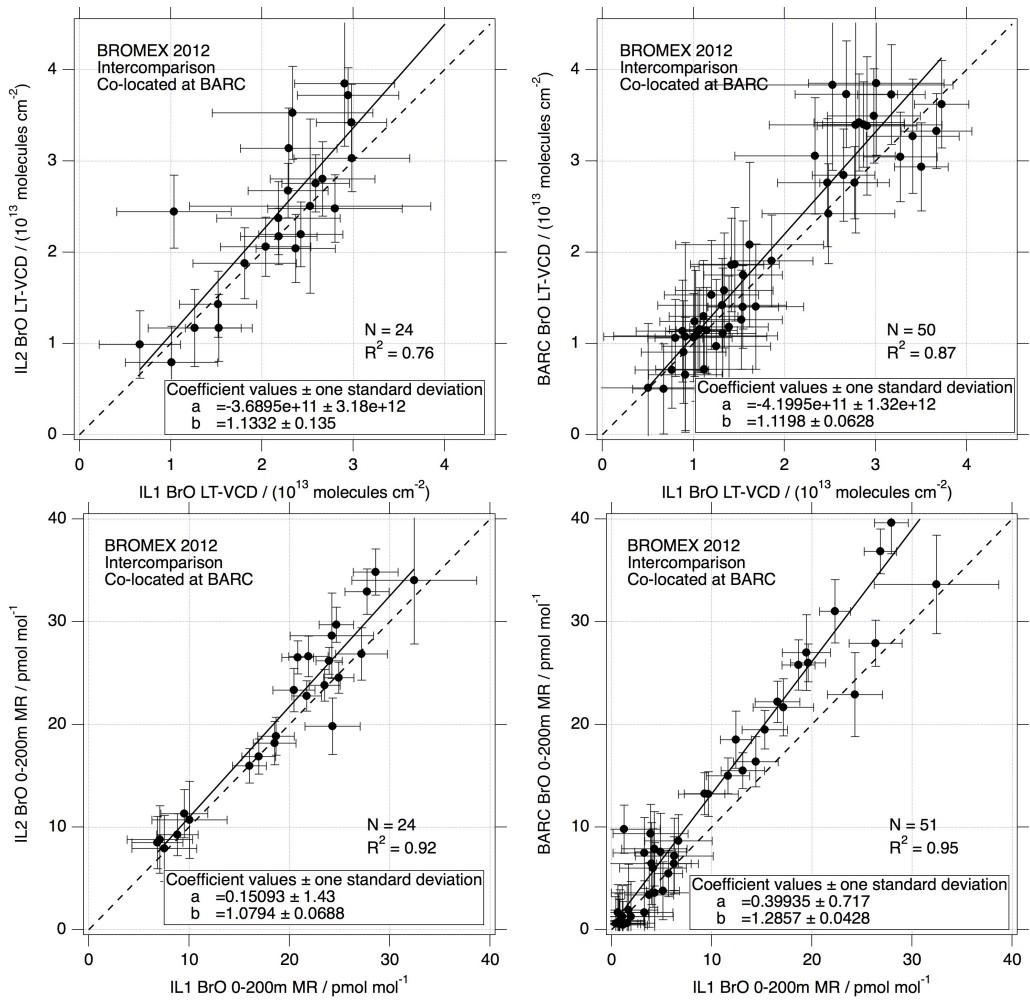

**Figure 3: Intercomparison of BrO measurements between BARC, IL1, and IL2 when all instruments were co-located at BARC.** Error bars (1σ) are shown on each data point, and were typically around 5 x $10^{12}$ molecule cm$^{-2}$ for BrO LT-VCD and 3 pmol mol$^{-1}$ for the BrO 0-200m mixing ratio (MR)





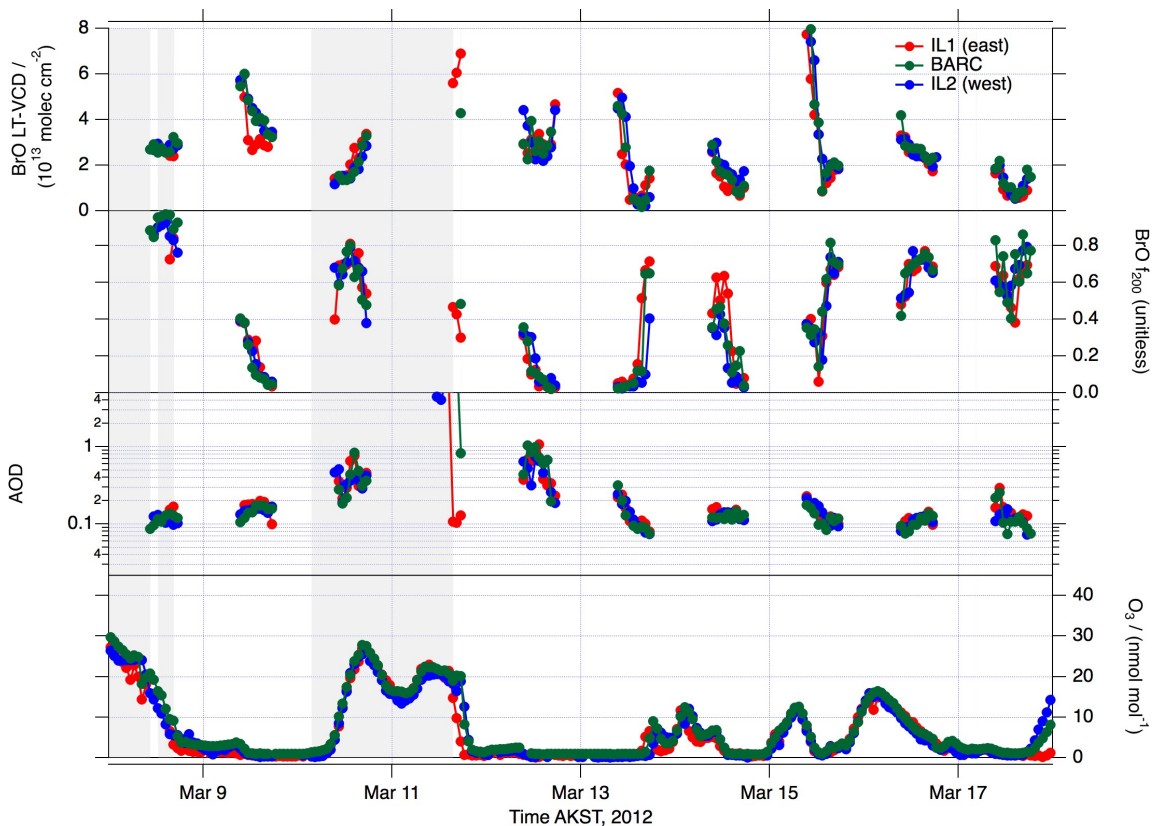

**Figure 4: Hourly BrO, aerosol optical depth (AOD), and ozone (O₃) data from each of the three sites (IL1, BARC, IL2) during
        period B1. The top panel shows the lower tropospheric VCD of BrO (integrated concentration from surface to top of model). The
        second panel shows the fraction of BrO LT-VCD in the lowest 200m, $f_{200}$ = VCD (0-200m) / LT-VCD. The third panel shows the
        vertical integral of the aerosol extinction from 0-4000m (the AOD). The lowest panel shows the in-situ surface ozone mixing ratio
        measured on the IL platforms and as measured by NOAA-GMD ~2 km northeast of the BARC building.**




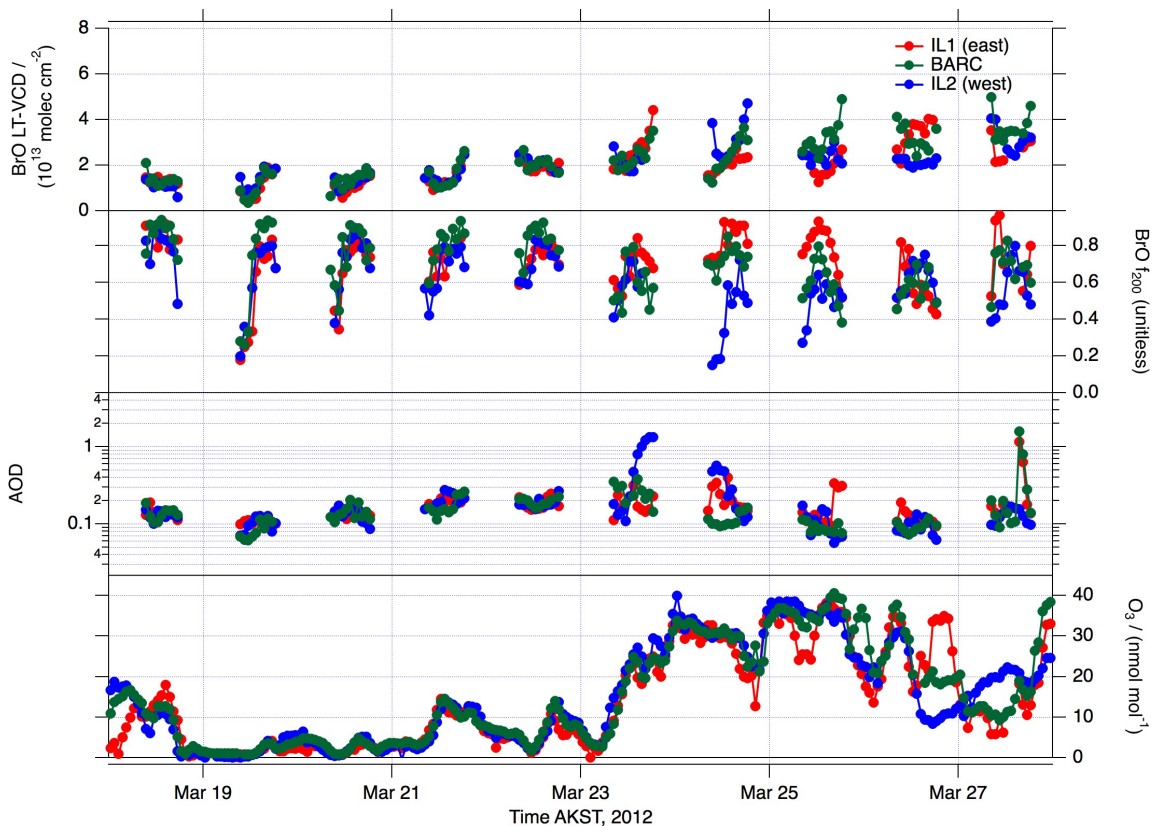

**Figure 5: The same as in Fig. 4, but for period B2. IL2 began drifting away from BARC on March 23.**





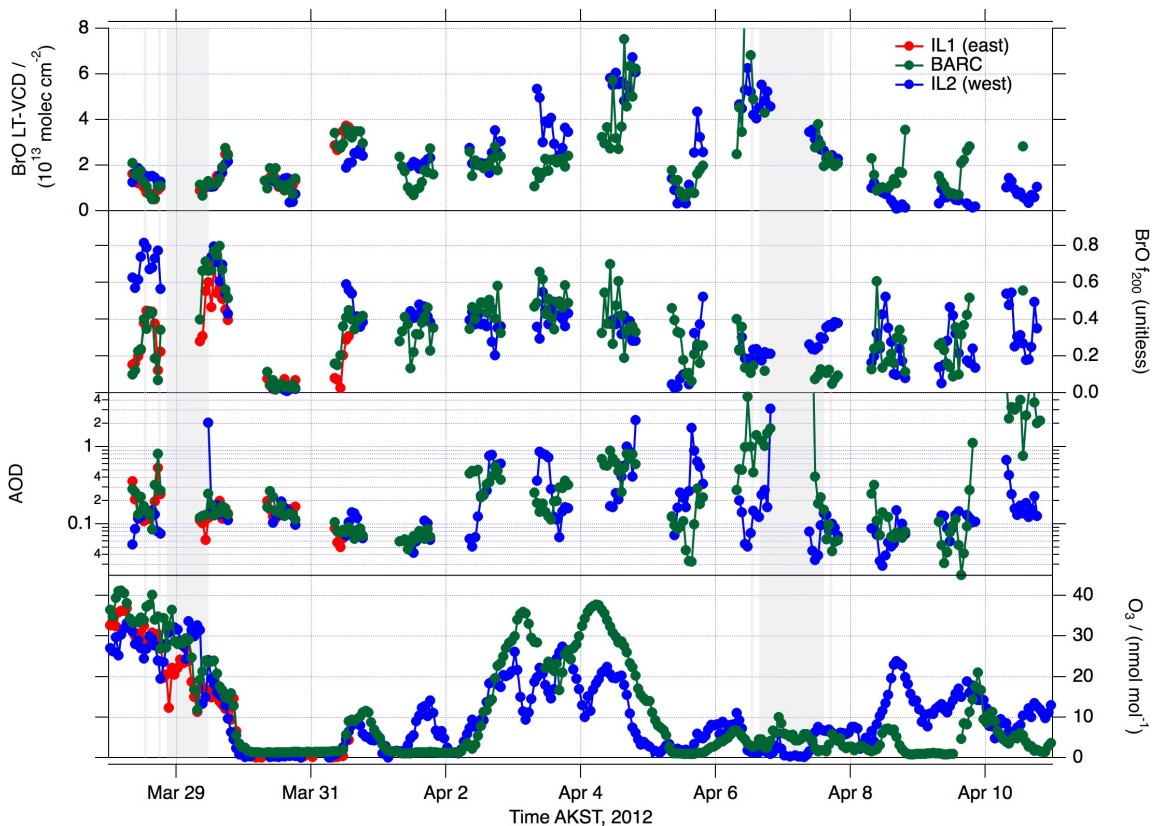

**Figure 6: The same as in Fig. 4, but for period B3. IL1 was recovered to BARC on March 27, and was co-located with the BARC instrument during this period, and was shut down at the end of March.**

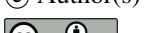



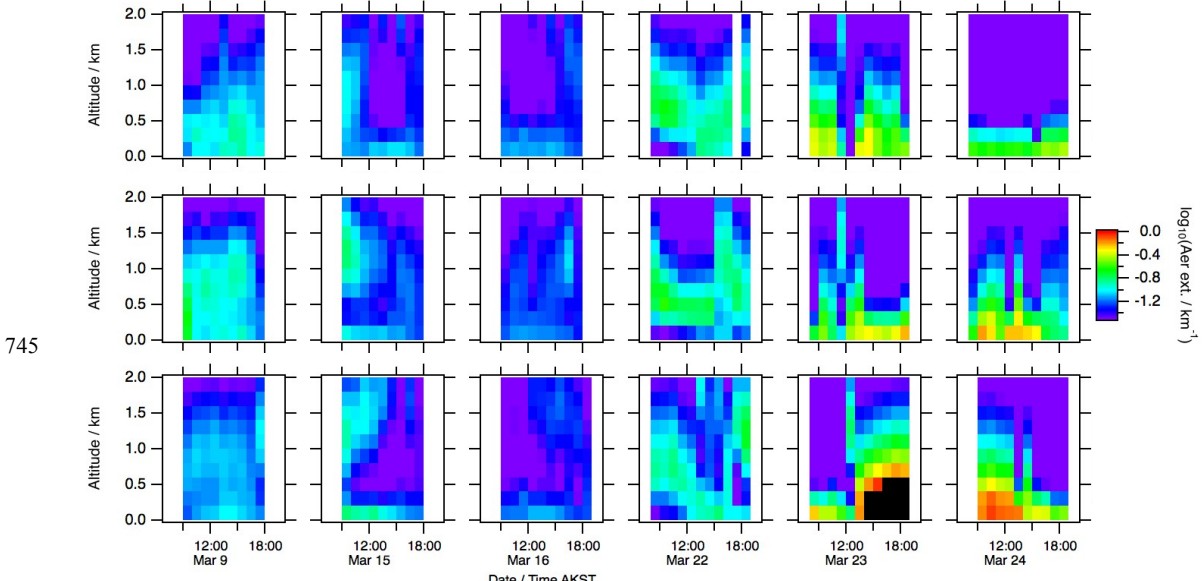

**Figure 7: Altitude/time profiles of aerosol particle extinction at three sites (top = IL1, mid = BARC, bottom = IL2) on selected days during BROMEX in 2012. Ticks on the time axis occur every three hours. Black pixels indicate extinction above 1 km$^{-1}$, which may be too optically thick to be reliably calculated via the optimal estimation analysis.**




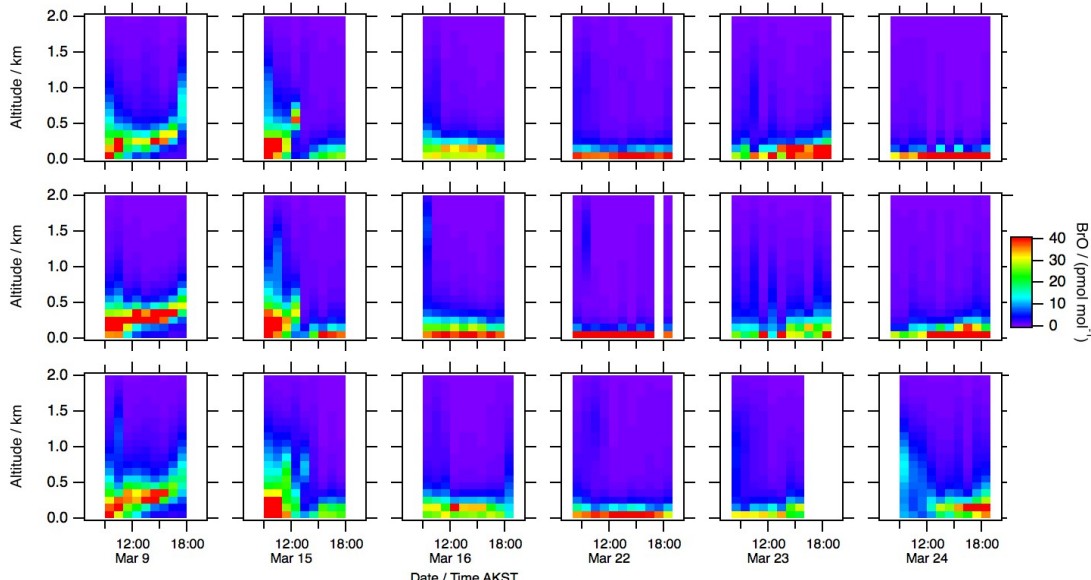

**Figure 8: Altitude/time profiles of BrO mixing ratio at three sites (top = IL1, mid = BARC, bottom = IL2) on selected days during BROMEX in 2012. Ticks on the time axis occur every three hours. BrO mixing ratios above 40 pmol mol⁻¹ are shown as black and are most likely artifacts of the limited vertical resolution of the optimal estimation analysis. White periods indicate missing data by either lack of sun, instrumental problems, or low visibility (e.g. afternoon of Mar 23 at IL2 due to the lead opening event).**





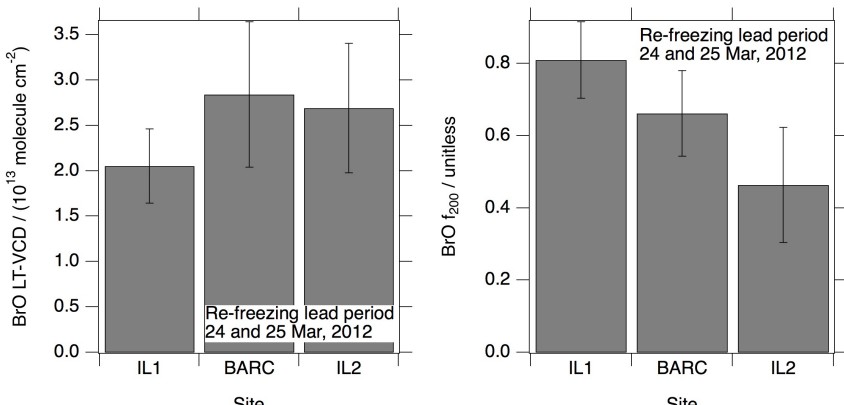

**Figure 9: Average BrO LT-VCD (left panel) and fraction BrO in lowest 200m, $f_{200}$, (right panel) during the two days after the lead opening, when the lead is open and re-freezing. The bar height is the average, and error bar length is +/- $1\sigma$. A t-test for significant difference ($\alpha = 0.05$) shows that the LT-VCD at BARC and IL2 was significantly larger than at IL1 but that BARC and IL2 were not significantly different. For the $f_{200}$ vertical distribution metric, all three sites were statistically different from each other with a clear trend from the upwind IL1 site to the downwind IL2 site.**
