# Peer review of "Horizontal and vertical structure of reactive bromine events probed by bromine monoxide MAX-DOAS spectroscopy"

_Atmospheric Chemistry and Physics, 2017_

## Referee Comment (RC1) · Anonymous Referee #1 · 3 Apr 2017

This paper presents data from 3 MAX-DOAS instruments deployed in 2012 as part of the BROMEX experiment. The paper comprises an important contribution to our knowledge of halogen activation and ozone depletion in the Arctic and should be accepted after some minor comments are addressed.

Minor comments:

Abstract, L32 – Please define more clearly sharp edge.

Figure 1 caption – Please define "cloud streets"

Figure 2 – This is the only figure that I cannot fully understand. In the 3rd panel, what are the flat red and blue lines? Were wind directions so consistent or is this

[Figure]

an instrument problem? Were their wind direction measurements on the IL1 and IL2 buoys? This is not clear for me.

L312, Section 5.2 – This section is titled "Snowpack-Based BrO events..." Does this refer to snow on sea-ice based events?

L337 – It would be nice to reference some additional work on bromine activation, including studies on aerosols at warmer temperatures. There is some evidence that there are interfacial/dark reactions that are also important. Two examples include:

- Hunt, S. W., et al. "Formation of molecular bromine from the reaction of ozone with deliquesced NaBr aerosol: evidence for interface chemistry." The Journal of Physical Chemistry A 108.52 (2004): 11559-11572.

- Oum, K. W., M. J. Lakin, and B. J. Finlayson-Pitts. "Bromine activation in the troposphere by the dark reaction of O3 with seawater ice." Geophysical Research Letters 25.21 (1998): 3923-3926.

L376 – There is a relevant study on HOBr uptake that should be mentioned here:

Roberts, Tjarda J., et al. "Re-evaluating the reactive uptake of HOBr in the troposphere with implications for the marine boundary layer and volcanic plumes." Atmospheric Chemistry and Physics 14.20 (2014): 11185-11199.

L393 – Sentence that starts with "As discussed by..." Please consider rephrasing as this sentence reads a bit strange.

L423 – I have two comments on this paragraph.

1. Can't other aerosols besides sea-salt also be formed/released from open leads? Is sea-salt the only potentially important aerosol surface that can be contributing here?

2. Aerosol extinction is lower at IL2 than the other sites. Can't this just be a limitation of the ability of the MAX-DOAS to measure aerosols aloft (higher than 1.5 km or so)? Given that the lead results in increased mixing, is it really there are less aerosols or are

aerosols just diluted and mixed out of the region where they can be measured reliably?

L452 – What does it mean to "deepen the BrO"?

L458 – Please rephrase the sentence that begins "Thus, in-situ. . ." to be more specific

L461 – The paper from Yang et al. (2010) and Theys et al. (2011) show good agreement with satellite observations in the Antarctic, but the model/satellite measurement comparison is less good in the Arctic. In addition, the study from Jones et al. (2009) primarily focuses on the Antarctic. Therefore, I think it's important to point out that the conclusion that "high winds" increase tropospheric BrO may be somewhat Antarctic specific. There may be different mechanisms that dominate in the Arctic because it's in general less stormy and more stable. I would add a reference to Theys et al. (2011) here as well:

Theys, N., et al. "Global observations of tropospheric BrO columns using GOME-2 satellite data." Atmospheric Chemistry and Physics 11.4 (2011): 1791.

Conclusions – I find it necessary to add a paragraph on what this means for future model studies/developments. Some examples of past work to mention include:

- Toyota, K., et al. "Air–snowpack exchange of bromine, ozone and mercury in the springtime Arctic simulated by the 1-D model PHANTAS–Part 1: In-snow bromine activation and its impact on ozone." Atmospheric Chemistry and Physics 14.8 (2014): 4101-4133.

- Toyota, K., et al. "Analysis of reactive bromine production and ozone depletion in the Arctic boundary layer using 3-D simulations with GEM-AQ: inference from synoptic-scale patterns." Atmospheric Chemistry and Physics 11.8 (2011): 3949.

- Holmes, Christopher D., Daniel J. Jacob, and Xin Yang. "Global lifetime of elemental mercury against oxidation by atomic bromine in the free troposphere." Geophysical Research Letters 33.20 (2006).

[Figure]

L506 – Missing period at the end of the paragraph.

Figures 4-6 – Consider using a different color than green so that it's easier to distinguish the green and blue curves. The dots are a bit big to see small differences in the measurements.

Figures 7 & 8 – Consider combining into one large paneled figure so that the BrO and aerosol profiles can be viewed together.

Figure 7 – Please comment in the text on the lower panel Mar 23 – What do the measurements mean above the black region? Are these real aerosol measurements or is everything above the black portion unreliable?

Figure 9 – Please mark on figure the upwind and downwind measurements for ease of understanding.

---

## Referee Comment (RC2) · Anonymous Referee #2 · 3 May 2017

[Summary]

In this study, Simpson et al. analyze the spatial and temporal evolution of BrO column densities in the lower troposphere and their vertical profiles as well as the aerosol optical depths and extinction profiles retrieved from three MAX-DOAS instruments deployed at and around Barrow (Utqiagvik) from early March to mid April 2012 during the BROMEX campaign. The analysis is complemented by the 250-m resolution MODIS satellite images of ice conditions along with data from in-situ measurements of surface ozone and meteorology collocated or nearly-collocated with the MAX-DOAS instruments. One of the three MAX-DOAS instruments was always located at the Barrow Arctic Research Center (BARC), whereas other two instruments (called the IceLanders

1 and 2) were sometimes collocated with the first instrument at BARC for the purpose of data quality assessment and at other times were deployed on the sea ice approximately 30-40 km to the east and west of BARC, respectively. At one point, there was an event of the lead opening across the sea ice near Barrow, when the IL2 started drifting further west up to about 250-km typically downwind from Barrow.

Several aspects that are important to the variability of BrO in the springtime Arctic lower troposphere are addressed in the present analysis: (1) the spatial scales of air masses containing the high levels of BrO in the absence of open and refrozen leads are sufficiently large so that the BrO distributions are quite homogeneous at the typical scales of satellite nadir-viewing pixel size except at the air-mass boundaries; (2) the prevalence of surface-bound shallow events of high BrO associated presumably with the release of gaseous bromine from the snowpack; (3) the lack of apparent impacts of open and refrozen leads as an immediate source of reactive bromine to the atmosphere but the vertical re-distributions of BrO due to enhanced mixing over the leads; (4) the repartitioning of BrO to other forms of bromine as a result of ozone depletion; and (5) the role of aerosol particles to sustain the high levels of BrO via heterogeneous reactions. The novel design of the field experiment, namely, spatial alignment of the MAX-DOAS instruments along the predominant wind directions around Barrow, has been executed generally well to show convincing cases, except for the role of the aerosol particles in the heterogeneous recycling of bromine, as I comment further below.

Overall, this study is no doubt an important contribution to the field. In my opinion, however, the argument related to the heterogeneous recycling of bromine remains speculative and requires further evidence (based either on additional field data or on trajectory/chemical-transport modeling) to characterize the air-mass history and the type of aerosols (especially whether they are sea salt or haze particles) detected optically by MAX-DOAS.

[Specific comments]

1. On the Line 221, Moore et al. (2014) is cited when the authors refer to the surface ozone recovery during the periods of higher wind speeds. If I understand correctly, Moore et al. (2014) is not an appropriate reference to cite in this context, because Moore et al. emphasize the role of convective mixing (thermal instability) over the leads rather than that of the turbulent mixing due to wind shear. I suggest the citation of Jacobi et al. (2010) and/or some other references that the authors see fit.

Jacobi, H.-W., et al.: Observation of widespread depletion of ozone in the springtime boundary layer of the central Arctic linked to mesoscale synoptic conditions, J. Geophys. Res., 115, D17302, doi:10.1029/2010JD013940, 2010.

2. The first half of the section 5.2 discusses the prevalence of surface-bound events of high BrO apparently associated with the release of gaseous bromine from the snowpack. Then, in the latter half of this subsection, the authors note the absence of high BrO aloft during the surface-bound BrO events on March 16 and 22, and seek the answers. On March 16, the MAX-DOAS detected not much aerosol extinction aloft and thus there would have been little chance to facilitate the heterogeneous recycling of bromine even if relatively high levels of total inorganic bromine were present. However, it is not clear to me whether the high levels of bromine should have existed aloft in the first place on this day. There is no discussion of the vertical profiles of atmospheric stability and air-mass history (backward trajectories, etc.). On March 22, the high aerosol extinctions were detected in the lofted layer from MAX-DOAS, whereas the increased levels of BrO were not observed aloft. The existence of strong boundary-layer temperature inversion was identified from meteorological sounding, pointing to the suppressed vertical mixing and decoupling of air masses between the surface and lofted layers. The authors seem to speculate either the lack of bromine sources or the predominance of non-acidic particles (which does not support the heterogeneous recycling of bromine) in the lofted layer on March 22. This case again seems to benefit from some discussion of air-mass history based on the backward trajectories, etc. According to Quinn et al. (2002), the chemical and optical properties of aerosols at Barrow are

strongly dependent on non-sea-salt sulfate during the spring. So it may well be that the high aerosol extinctions observed aloft on March 22 were associated with the anthropogenic haze particles and that the lofted air mass was virtually devoid of bromine due the lack of recent contact with saline ice surfaces and/or sea-salt aerosols.

Quinn, P. K., T. L. Miller, T. S. Bates, J. A. Ogren, E. Andrews, and G. E Shaw, A 3-year record of simultaneously measured aerosol chemical and optical properties at Barrow, Alaska, J. Geophys. Res., 107(D11), doi:10.1029/2001JD001248, 2002.

3. Section 5.3 discusses the impact of decreasing ozone concentrations on the re-partitioning of BrO to other forms of bromine, perhaps Br-atoms, HBr and particulate bromide. It seems useful to refer to the results from photochemical modeling studies (e.g., Sander et al., 1997; Evans et al., 2003; Toyota et al., 2014) which are generally consistent with the present finding.

Sander, R., et al.: Modeling the chemistry of ozone, halogen compounds and hydro-carbons in the arctic troposphere during spring, Tellus Ser. B, 49, 522-532, 1997.

Evans, M. J., et al., Coupled evolution of BrOx-ClOx-HOx-NOx chemistry during bromine-catalyzed ozone depletion events in the arctic boundary layer, J. Geophys. Res., 108(D4), 8368, doi:10.1029/2002JD002732, 2003.

Toyota, K., et al.: Air–snowpack exchange of bromine, ozone and mercury in the spring-time Arctic simulated by the 1-D model PHANTAS – Part 1: In-snow bromine activation and its impact on ozone, Atmos. Chem. Phys., 14, 4101-4133, doi:10.5194/acp-14-4101-2014, 2014.

4. Section 5.4 digs the role of heterogeneous recycling of bromine in/on the aerosol particles by estimating the rate of reactive uptake of HOBr on the aerosols based on the aerosol extinctions measured by MAX-DOAS. The discussion provided there partly answers the questions raised in section 5.2 as to why the presence of a certain amount of aerosols is required for sustaining the high levels of BrO. I would have liked this subsection better if the authors had attempted some photochemical box model simulations to back up their argument further. As it is probably too demanding to request the completion of this task within the time frame of the manuscript revision, I suggest the authors to state that the task is remaining for modelers to back up what the authors speculate in this study. On the other hand, the discussion related to the absence of high BrO levels aloft in the presence of high aerosol extinctions remains speculative and does not really offer anything conclusive. This subsection sounds rather indecisive overall and I find it the weakness of this study. There are a few minor points that I would like the authors to consider. First, the authors use Qext = 4 to convert the aerosol extinction to the aerosol surface area, but I wonder if Qext = 2 is a more representative asymptotic value for this calculation. Second, the authors derive the important threshold value, namely, aerosol extinction > 0.1 km-1, as a requirement for BrO to exist aloft. I think this threshold value should be referred to in the abstract as well. Third, if I remember correctly, Wachsmuth et al. (2002) investigated the gamma(HOBr) on sea salt, hence the authors should state this more clearly and note that gamma = 0.6 is probably an upper limit.

5. Section 5.5 discusses an interesting case of the lead opening and subsequent refreezing event. There was not a significant increase in the total BrO column densities in the lower troposphere downwind of the leads (indicating the lack of strong bromine sources affecting the level of bromine on the time scale of hours), whereas there were obvious changes in the vertical BrO profiles due to enhanced vertical mixing. There are multiple factors that can affect the BrO column densities and their profiles over and downwind of the open and refrozen leads, which I think are generally discussed/covered by the authors with appropriate references. One additional point that I would like the authors to note is the potential role of (super-cooled) liquid cloud water associated with the open leads in the suppression of reactive bromine chemistry as discussed by Piot and von Glasow (2008). Another relatively minor note is that, on the second paragraph of this subsection, the authors use the wind speed (ca. 5 m/s) in the surface boundary layer to estimate the time scale of transport of air between

the MAX-DOAS deployment sites, which could be revised by using the wind speeds (perhaps greater than 5 m/s) obtained from meteorological sounding at Barrow and relevant to the deeper layer of interest (up to 1 km AGL) in this discussion of air mass transport.

Piot, M. and von Glasow, R.: The potential importance of frost flowers, recycling on snow, and open leads for ozone depletion events, Atmos. Chem. Phys., 8, 2437-2467, doi:10.5194/acp-8-2437-2008, 2008.

6. Section 5.6 synthesizes the findings and discussions from the previous subsections with prior studies. As noted above, I feel that the discussion related to the heterogeneous recycling of bromine on the aerosols remains inconclusive and therefore the latter half of the statement in the starting sentence of section 5.6 is not fully supported (i.e., "... may over time increase the column density of BrO through heterogeneous chemistry on lofted aerosol particles."). I would like the authors either to revise the content of the paper significantly to make this first sentence more compelling or to revise this sentence itself. Otherwise, I find this subsection interesting. A minor point, but I would like the authors to state more explicitly what they mean by "chemical composition, which is not conducive to reactive bromine production" (Lines 465-466). Also, Toyota et al. (2011) could be cited along with Jones et al. (2009), Begoin et al. (2010) and Choi et al. (2012) when referring to the prior studies reporting the role of mesoscale cyclonic storms and high winds in the occurrence of high BrO column densities.

Toyota, K., et al.: Analysis of reactive bromine production and ozone depletion in the Arctic boundary layer using 3-D simulations with GEM-AQ: inference from synoptic-scale patterns, Atmos. Chem. Phys., 11, 3949-3979, doi:10.5194/acp-11-3949-2011, 2011.

7. Although the manuscript is generally well written, the discussion section (Section 5) appears to benefit from another round of careful editing by the authors to improve some of the wording beyond what I suggest below.

[Technical suggestions]

L41-42: Fix the location of commas around the citation to references.

L43: pollution -> pollutants

L45: lacking -> very limited

L133: ozone limit of detection -> detection limit of ozone

L155-156: Winds/winds -> Wind speeds/wind speeds

L240: more variability -> notable discrepancy

L240: appear similar -> vary similarly

L256: presence -> occurrence

L257: BrO -> BrO aloft

L262: . . ., which decreased to lower values, . . .

L274: more shallow than March 16 -> shallower than on March 16

L288: gradients -> spatial gradients

L299: correlations -> column densities

L307: lengths -> length

L397: Delete the comma after "Peterson et al. (2017)".

L409: even -> event

L423: lead -> leads

L430-431: . . ., so all of the submicron aerosol particles, supermicron particles and solid/liquid water droplets . . .

L436: and most downwind site -> especially at the most downwind site

[Figure]

L452: These data show that vertical mixing deepens the atmospheric layer containing BrO through . . .

L459-461: This sentence sounds a bit awkward to me. Consider rephrasing.

L483: showed -> gave

L483: Change the colon (:) to the period (.).

L710 (Fig. 1 caption): streets -> streaks

Figs.7-8: Add legends in the plots to indicate that the top, middle and bottom panels correspond to data from IL1, BARC and IL2, respectively.

---

## Author Comment (AC1) · 12 Jun 2017

**We thank Reviewer 1 for helpful comments. In this response, the reviewer comments are included in plain text and our responses are in bold text. Line numbers refer to the ACPD version of the manuscript.**

This paper presents data from 3 MAX-DOAS instruments deployed in 2012 as part of the BROMEX experiment. The paper comprises an important contribution to our knowledge of halogen activation and ozone depletion in the Arctic and should be accepted after some minor comments are addressed.

[Figure]

**We appreciate the positive words on the manuscript.**

Minor comments:

Abstract, L32 – Please define more clearly sharp edge. Figure 1 caption – Please define "cloud streets"

**Wording was clarified as "sharp" meaning "smaller than 30km horizontal length scale". Cloud streets are horizontal convective rolls associated with airmass motion from over ice to over open water.**

Figure 2 – This is the only figure that I cannot fully understand. In the 3rd panel, what are the flat red and blue lines? Were wind directions so consistent or is this an instrument problem? Were their wind direction measurements on the IL1 and IL2 buoys? This is not clear for me.

**These red and blue lines represent the direction between the IL1 and IL2 buoys from the BARC site, which was not made clear in the text, so we modified the caption to be more informative.**

L312, Section 5.2 – This section is titled "Snowpack-Based BrO events. . ." Does this refer to snow on sea-ice based events?

**Both sea ice and land at this time of year are covered with snowpack, and past literature has demonstrated that "shallow" (large fraction of BrO in the lowest 200m) events are associated with recycling of reactive halogens on snowpack surfaces. For the most part, these events come from sea ice regions, but we do not explicitly separate snow on sea ice from snow on land.**

L337 – It would be nice to reference some additional work on bromine activation, including studies on aerosols at warmer temperatures. There is some evidence that there are interfacial/dark reactions that are also important. Two examples include:

- Hunt, S. W., et al. "Formation of molecular bromine from the reaction of ozone with

deliquesced NaBr aerosol: evidence for interface chemistry." The Journal of Physical Chemistry A 108.52 (2004): 11559-11572.

- Oum, K. W., M. J. Lakin, and B. J. Finlayson-Pitts. "Bromine activation in the troposphere by the dark reaction of O3 with seawater ice." Geophysical Research Letters 25.21 (1998): 3923-3926.

**These citations are useful and have been added, but the section at line 337 is specifically about the pH dependence of reaction 3, so we didn't add them at line 337, but instead at line 324 . Note that these processes are also heterogeneous processes, requiring either ice (snowpack) or aerosol surfaces, so they are similar to reaction 3 listed.**

L376 – There is a relevant study on HOBr uptake that should be mentioned here: Roberts, Tjarda J., et al. "Re-evaluating the reactive uptake of HOBr in the troposphere with implications for the marine boundary layer and volcanic plumes." Atmospheric Chemistry and Physics 14.20 (2014): 11185-11199.

**The reference was added.**

L393 – Sentence that starts with "As discussed by. . ." Please consider rephrasing as this sentence reads a bit strange.

**We reworded this sentence.**

L423 – I have two comments on this paragraph.

1. Can't other aerosols besides sea-salt also be formed/released from open leads? Is sea-salt the only potentially important aerosol surface that can be contributing here?

2. Aerosol extinction is lower at IL2 than the other sites. Can't this just be a limitation of the ability of the MAX-DOAS to measure aerosols aloft (higher than 1.5 km or so)? Given that the lead results in increased mixing, is it really there are less aerosols or are aerosols just diluted and mixed out of the region where they can be measured reliably?

**Yes, potentially other particles could be released from an open lead. For example, particles derived from the sea surface microlayer could be produced. Thus, the wording was broadened. The reduced aerosol extinction at the downwind site is not likely due to technical problems with MAX-DOAS in this case. Specifically, the aerosol extinction profiles (Fig. 7) show that the aerosol is not highly lofted. Additionally, as AOD becomes lower (e.g. the better the visibility), MAX-DOAS becomes more able to see higher in the atmosphere.**

L452 – What does it mean to "deepen the BrO"?

**We clarified "BrO layer thickness"**

L458 – Please rephrase the sentence that begins "Thus, in-situ. . ." to be more specific

**Clarified to mean surface observations (e.g. ground-based CIMS or MAX-DOAS)**

L461 – The paper from Yang et al. (2010) and Theys et al. (2011) show good agreement with satellite observations in the Antarctic, but the model/satellite measurement comparison is less good in the Arctic. In addition, the study from Jones et al. (2009) primarily focuses on the Antarctic. Therefore, I think it's important to point out that the conclusion that "high winds" increase tropospheric BrO may be somewhat Antarctic specific. There may be different mechanisms that dominate in the Arctic because it's in general less stormy and more stable. I would add a reference to Theys et al. (2011) here as well:

Theys, N., et al. "Global observations of tropospheric BrO columns using GOME-2 satellite data." Atmospheric Chemistry and Physics 11.4 (2011): 1791.

**This bi-polar difference in wind speed regime was added to the section.**

Conclusions – I find it necessary to add a paragraph on what this means for future model studies/developments. Some examples of past work to mention include:

- Toyota, K., et al. "Air–snowpack exchange of bromine, ozone and mercury in the

springtime Arctic simulated by the 1-D model PHANTAS–Part 1: In-snow bromine activation and its impact on ozone." Atmospheric Chemistry and Physics 14.8 (2014): 4101-4133.

- Toyota, K., et al. "Analysis of reactive bromine production and ozone depletion in the Arctic boundary layer using 3-D simulations with GEM-AQ: inference from synoptic-scale patterns." Atmospheric Chemistry and Physics 11.8 (2011): 3949.

- Holmes, Christopher D., Daniel J. Jacob, and Xin Yang. "Global lifetime of elemental mercury against oxidation by atomic bromine in the free troposphere." Geophysical Research Letters 33.20 (2006).

**Certainly, there is a need for modeling of these data; these citations and other modeling efforts were added.**

L506 – Missing period at the end of the paragraph.

**Fixed.**

Figures 4-6 – Consider using a different color than green so that it's easier to distinguish the green and blue curves. The dots are a bit big to see small differences in the measurements.

**The green was chosen as a progression from red-green-blue, so we didn't change that color. Instead, we changed the symbol for the green lines to have an open symbol for better differentiation from the blue line. We reduced the size of the dots to allow small differences to be seen.**

Figures 7  8 – Consider combining into one large paneled figure so that the BrO and aerosol profiles can be viewed together.

**We combined these figures to make a single very large one (new Fig. 7).**

Figure 7 – Please comment in the text on the lower panel Mar 23 – What do the measurements mean above the black region? Are these real aerosol measurements

or is everything above the black portion unreliable?

**This is a good point. The aerosol measurements above an optically thick layer are not reliable (you cannot see through the thick layer). The text was modified.**

Figure 9 – Please mark on figure the upwind and downwind measurements for ease of understanding.

**Upwind and downwind are marked.**

—————————————————————

---

## Author Comment (AC2) · 12 Jun 2017

**We thank the second reviewer for helpful comments. In this response, the reviewer comments are included in plain text and our responses are in bold text. Line numbers refer to the ACPD version of the manuscript.**

In this study, Simpson et al. analyze the spatial and temporal evolution of BrO column densities in the lower troposphere and their vertical profiles as well as the aerosol optical depths and extinction profiles retrieved from three MAX-DOAS instruments deployed at and around Barrow (Utqiagvik) from early March to mid April 2012 during the BROMEX campaign. The analysis is complemented by the 250-m resolution MODIS

satellite images of ice conditions along with data from in-situ measurements of surface ozone and meteorology collocated or nearly-collocated with the MAX-DOAS instruments. One of the three MAX-DOAS instruments was always located at the Barrow Arctic Research Center (BARC), whereas other two instruments (called the IceLanders 1 and 2) were sometimes collocated with the first instrument at BARC for the purpose of data quality assessment and at other times were deployed on the sea ice approximately 30-40 km to the east and west of BARC, respectively. At one point, there was an event of the lead opening across the sea ice near Barrow, when the IL2 started drifting further west up to about 250-km typically downwind from Barrow. Several aspects that are important to the variability of BrO in the springtime Arctic lower troposphere are addressed in the present analysis: (1) the spatial scales of air masses containing the high levels of BrO in the absence of open and refrozen leads are sufficiently large so that the BrO distributions are quite homogeneous at the typical scales of satellite nadir-viewing pixel size except at the air-mass boundaries; (2) the prevalence of surface-bound shallow events of high BrO associated presumably with the release of gaseous bromine from the snowpack; (3) the lack of apparent impacts of open and refrozen leads as an immediate source of reactive bromine to the atmosphere but the vertical re-distributions of BrO due to enhanced mixing over the leads; (4) the repartitioning of BrO to other forms of bromine as a result of ozone depletion; and (5) the role of aerosol particles to sustain the high levels of BrO via heterogeneous reactions. The novel design of the field experiment, namely, spatial alignment of the MAX-DOAS instruments along the predominant wind directions around Barrow, has been executed generally well to show convincing cases, except for the role of the aerosol particles in the heterogeneous recycling of bromine, as I comment further below. Overall, this study is no doubt an important contribution to the field. In my opinion, however, the argument related to the heterogeneous recycling of bromine remains speculative and requires further evidence (based either on additional field data or on trajectory/chemical-transport modeling) to characterize the air-mass history and the type of aerosols (especially whether they are sea salt or haze particles) detected optically by MAX-DOAS.

**This manuscript reports the data from this field study. We see this as the starting point for future modeling efforts, as suggested by the reviewer. As the reviewer points out, these MAX-DOAS aerosol observations are of optical properties while the chemical nature of the aerosol is what is important for halogen activation. In select cases (e.g. the aircraft observations described in Peterson et al. (2017) in press, discussion available at https://doi.org/10.5194/acp-2016-1141), we have more information, which is consistent with recycling of BrO aloft on aerosol surfaces. However, the present manuscript describes the full BROMEX field campaign, a period for which lofted aerosol chemical data is not generally available. Therefore, we have assured that the wording of this manuscript indicates that the data are "consistent with" recycling on aerosol surfaces. We also feel that the data presented here is an excellent case study for future chemical-transport modeling efforts, which can address this idea further. We have added discussion encouraging future modeling efforts.**

[Specific comments]

1. On the Line 221, Moore et al. (2014) is cited when the authors refer to the surface ozone recovery during the periods of higher wind speeds. If I understand correctly, Moore et al. (2014) is not an appropriate reference to cite in this context, because Moore et al. emphasize the role of convective mixing (thermal instability) over the leads rather than that of the turbulent mixing due to wind shear. I suggest the citation of Jacobi et al. (2010) and/or some other references that the authors see fit. Jacobi, H.-W., et al.: Observation of widespread depletion of ozone in the springtime boundary layer of the central Arctic linked to mesoscale synoptic conditions, J. Geophys. Res., 115, D17302, doi:10.1029/2010JD013940, 2010.

**This is a better citation for wind-induced vertical mixing, which we have substituted for the prior reference to Moore et al. (2014).**

2. The first half of the section 5.2 discusses the prevalence of surface-bound events of

high BrO apparently associated with the release of gaseous bromine from the snow-pack. Then, in the latter half of this subsection, the authors note the absence of high BrO aloft during the surface-bound BrO events on March 16 and 22, and seek the answers. On March 16, the MAX-DOAS detected not much aerosol extinction aloft and thus there would have been little chance to facilitate the heterogeneous recycling of bromine even if relatively high levels of total inorganic bromine were present. However, it is not clear to me whether the high levels of bromine should have existed aloft in the first place on this day. There is no discussion of the vertical profiles of atmospheric stability and air-mass history (backward trajectories, etc.). On March 22, the high aerosol extinctions were detected in the lofted layer from MAX-DOAS, whereas the increased levels of BrO were not observed aloft. The existence of strong boundary-layer temperature inversion was identified from meteorological sounding, pointing to the suppressed vertical mixing and decoupling of air masses between the surface and lofted layers. The authors seem to speculate either the lack of bromine sources or the predominance of non-acidic particles (which does not support the heterogeneous recycling of bromine) in the lofted layer on March 22. This case again seems to benefit from some discussion of air-mass history based on the backward trajectories, etc. According to Quinn et al. (2002), the chemical and optical properties of aerosols at Barrow are strongly dependent on non-sea-salt sulfate during the spring. So it may well be that the high aerosol extinctions observed aloft on March 22 were associated with the anthropogenic haze particles and that the lofted air mass was virtually devoid of bromine due the lack of recent contact with saline ice surfaces and/or sea-salt aerosols.

Quinn, P. K., T. L. Miller, T. S. Bates, J. A. Ogren, E. Andrews, and G. E Shaw, A 3-year record of simultaneously measured aerosol chemical and optical properties at Barrow, Alaska, J. Geophys. Res., 107(D11), doi:10.1029/2001JD001248, 2002.

**It is quite possible that the aerosol extinction aloft on March 22 is anthropogenic haze, but the MAX-DOAS optical observations unfortunately do not distinguish between haze and sea salt. Moore et al. (2014) extensively studied trajectories**

**during this period (see Fig. 2) for these trajectories. During this period (March 22-25), airmasses come from the sea ice (from NE through ENE), thus if they were influenced by anthropogenic sulfate pollution, it would have been days earlier, on which timescale the trajectory calculation may be in error and/or mixing of airmasses could have destroyed the integrity of the air parcel. Therefore, it would be hard to say where the optically detected extinction aloft came from definitively. For this reason we discuss possible reasons for the observation that BrO is not aloft given the presence of aerosol extinction aloft. We encourage future work into this question, and we cited the Quinn et al. (2002) paper in the discussion of this point.**

3. Section 5.3 discusses the impact of decreasing ozone concentrations on the repartitioning of BrO to other forms of bromine, perhaps Br-atoms, HBr and particulate bromide. It seems useful to refer to the results from photochemical modeling studies (e.g., Sander et al., 1997; Evans et al., 2003; Toyota et al., 2014) which are generally consistent with the present finding.

Sander, R., et al.: Modeling the chemistry of ozone, halogen compounds and hydrocarbons in the arctic troposphere during spring, Tellus Ser. B, 49, 522-532, 1997.

Evans, M. J., et al., Coupled evolution of BrOx-ClOx-HOx-NOx chemistry during bromine-catalyzed ozone depletion events in the arctic boundary layer, J. Geophys. Res., 108(D4), 8368, doi:10.1029/2002JD002732, 2003.

Toyota, K., et al.: Air–snowpack exchange of bromine, ozone and mercury in the springtime Arctic simulated by the 1-D model PHANTAS – Part 1: In-snow bromine activation and its impact on ozone, Atmos. Chem. Phys., 14, 4101-4133, doi:10.5194/acp-144101-2014, 2014.

**It is a very good point that models also show BrOx repartitioning. We added discussion of model results to the introduction where this topic is discussed and to this discussion.**

4. Section 5.4 digs the role of heterogeneous recycling of bromine in/on the aerosol particles by estimating the rate of reactive uptake of HOBr on the aerosols based on the aerosol extinctions measured by MAX-DOAS. The discussion provided there partly answers the questions raised in section 5.2 as to why the presence of a certain amount of aerosols is required for sustaining the high levels of BrO. I would have liked this subsection better if the authors had attempted some photochemical box model simulations to back up their argument further. As it is probably too demanding to request the completion of this task within the time frame of the manuscript revision, I suggest the authors to state that the task is remaining for modelers to back up what the authors speculate in this study. On the other hand, the discussion related to the absence of high BrO levels aloft in the presence of high aerosol extinctions remains speculative and does not really offer anything conclusive. This subsection sounds rather indecisive overall and I find it the weakness of this study. There are a few minor points that I would like the authors to consider. First, the authors use Qext = 4 to convert the aerosol extinction to the aerosol surface area, but I wonder if Qext = 2 is a more representative asymptotic value for this calculation. Second, the authors derive the important threshold value, namely, aerosol extinction > 0.1 km-1, as a requirement for BrO to exist aloft. I think this threshold value should be referred to in the abstract as well. Third, if I remember correctly, Wachsmuth et al. (2002) investigated the gamma(HOBr) on sea salt, hence the authors should state this more clearly and note that gamma = 0.6 is probably an upper limit.

**As stated earlier, we encourage future photochemical modeling of the data described in this manuscript. These estimates are simply provided so that we can show that the observed aerosol extinction, when approximately converted to a surface area density, gives heterogeneous chemical rates that are on the appropriate timescale. For this purpose, we made the choice of Qext = 4 to be most conservative (e.g. give the lowest estimate) with respect to how much surface area density would be necessary to give the observed aerosol extinction. It is agreed, and we modified the text to express the idea that gamma=0.6 is likely an**

**upper limit. The true result of this study is the optical extinction threshold, so this was added to the abstract.**

5. Section 5.5 discusses an interesting case of the lead opening and subsequent refreezing event. There was not a significant increase in the total BrO column densities in the lower troposphere downwind of the leads (indicating the lack of strong bromine sources affecting the level of bromine on the time scale of hours), whereas there were obvious changes in the vertical BrO profiles due to enhanced vertical mixing. There are multiple factors that can affect the BrO column densities and their profiles over and downwind of the open and refrozen leads, which I think are generally discussed/covered by the authors with appropriate references. One additional point that I would like the authors to note is the potential role of (super-cooled) liquid cloud water associated with the open leads in the suppression of reactive bromine chemistry as discussed by Piot and von Glasow (2008). Another relatively minor note is that, on the second paragraph of this subsection, the authors use the wind speed (ca. 5 m/s) in the surface boundary layer to estimate the time scale of transport of air between the MAX-DOAS deployment sites, which could be revised by using the wind speeds (perhaps greater than 5 m/s) obtained from meteorological sounding at Barrow and relevant to the deeper layer of interest (up to 1 km AGL) in this discussion of air mass transport.

Piot, M. and von Glasow, R.: The potential importance of frost flowers, recycling on snow, and open leads for ozone depletion events, Atmos. Chem. Phys., 8, 2437-2467, doi:10.5194/acp-8-2437-2008, 2008.

**We examined the wind speed on both days after the lead opening, and find that the wind speed increases aloft, as expected. The aerosol particles and BrO on these days are primarily in the lowest 600m, so we averaged the wind speed in the 0-600m AGL region. We find that the daytime soundings (15 AKST) on 24 and 25 Mar, 2012 had average windspeeds of 9 and 8 m/s, respectively, so we now use 8.5 m / s as the wind speed. We added text related to the Piot and von**

**Glasow modeling study. The MODIS images show that there is not cloud at IL2, but there appears to be thin cloud between BARC and IL2, so this idea may be relevant to the lack of increased reactive halogens downwind of the re-freezing lead.**

6. Section 5.6 synthesizes the findings and discussions from the previous subsections with prior studies. As noted above, I feel that the discussion related to the heterogeneous recycling of bromine on the aerosols remains inconclusive and therefore the latter half of the statement in the starting sentence of section 5.6 is not fully supported (i.e., ". . . may over time increase the column density of BrO through heterogeneous chemistry on lofted aerosol particles."). I would like the authors either to revise the content of the paper significantly to make this first sentence more compelling or to revise this sentence itself. Otherwise, I find this subsection interesting. A minor point, but I would like the authors to state more explicitly what they mean by "chemical composition, which is not conducive to reactive bromine production" (Lines 465-466). Also, Toyota et al. (2011) could be cited along with Jones et al. (2009), Begoin et al. (2010) and Choi et al. (2012) when referring to the prior studies reporting the role of mesoscale cyclonic storms and high winds in the occurrence of high BrO column densities. Toyota, K., et al.: Analysis of reactive bromine production and ozone depletion in the Arctic boundary layer using 3-D simulations with GEM-AQ: inference from synoptic-scale patterns, Atmos. Chem. Phys., 11, 3949-3979, doi:10.5194/acp-11-3949-2011, 2011.

**We agree that the vertical redistribution of BrO is the largest observed feature and the column increase is a smaller aspect, so we adjusted the wording as suggested by the reviewer. The comment on "which is not conductive to" has been changed to "which has incorrect pH for". The Toyota reference was added.**

7. Although the manuscript is generally well written, the discussion section (Section 5) appears to benefit from another round of careful editing by the authors to improve some of the wording beyond what I suggest below.

**Thank you for these suggested wording improvements. We addressed them all, with a few additional comments below.**

[Technical suggestions]

L41-42: Fix the location of commas around the citation to references.

L43: pollution -> pollutants

L45: lacking -> very limited

L133: ozone limit of detection -> detection limit of ozone

L155-156: Winds/winds -> Wind speeds/wind speeds

L240: more variability -> notable discrepancy

L240: appear similar -> vary similarly

L256: presence -> occurrence

L257: BrO -> BrO aloft

L262: . . ., which decreased to lower values, . . .

L274: more shallow than March 16 -> shallower than on March 16

L288: gradients -> spatial gradients

L299: correlations -> column densities

L307: lengths -> length

L397: Delete the comma after "Peterson et al. (2017)".

L409: even -> event

L423: lead -> leads

L430-431: . . ., so all of the submicron aerosol particles, supermicron particles and

solid/liquid water droplets . . .

L436: and most downwind site -> especially at the most downwind site

L452: These data show that vertical mixing deepens the atmospheric layer containing BrO through . . .

L459-461: This sentence sounds a bit awkward to me. Consider rephrasing.

L483: showed -> gave

L483: Change the colon (:) to the period (.).

L710 (Fig. 1 caption): streets -> streaks

**This point was also mentioned by the other reviewer. The term "cloud streets" is a technical term for horizontal convective rolls associated with airmass motion from over ice to over open water. Thus, we kept the term, but clarified it.**

Figs.7-8: Add legends in the plots to indicate that the top, middle and bottom panels correspond to data from IL1, BARC and IL2, respectively.

**Based upon input from the other reviewer, we combined these figures and added legends to clarify the sites in each row.**
* * *

---

## Author Response (AR1)

Hello Editor Anna Jones-

 We have addressed the reviewer comments via the ACPD online commenting system. Please see the online discussion for our replies.  The revised manuscript printed in "tracked changes" mode follows this cover page.  Please note that the tracked changes mode is not terribly clear about deletion / replacement of figures, but the final figures are visible on the resubmission version.

 Best regards,

William (Bill) Simpson

[revised manuscript text omitted]